# Sharpness-Aware Minimization Enhances Feature Quality via Balanced Learning

**Jacob Mitchell Springer[1], Vaishnavh Nagarajan[2] & Aditi Raghunathan[1]**
[1]Carnegie Mellon University    [2]Google Research
{jspringer,raditi}@cmu.edu[1] vaishnavh@google.com[2]

## Abstract

Sharpness-Aware Minimization (SAM) has emerged as a promising alternative optimizer to stochastic gradient descent (SGD). The originally-proposed motivation behind SAM was to bias neural networks towards flatter minima that are believed to generalize better. However, recent studies have shown conflicting evidence on the relationship between flatness and generalization, suggesting that flatness does fully explain SAM's success. Sidestepping this debate, we identify an orthogonal effect of SAM that is beneficial out-of-distribution: we argue that SAM implicitly balances the quality of diverse features. SAM achieves this effect by adaptively suppressing well-learned features which gives remaining features opportunity to be learned. We show that this mechanism is beneficial in datasets that contain redundant or spurious features where SGD falls for the simplicity bias and would not otherwise learn all available features. Our insights are supported by experiments on real data: we demonstrate that SAM improves the quality of features in datasets containing redundant or spurious features, including CelebA, Waterbirds, CIFAR-MNIST, and DomainBed.

## 1 Introduction

Sharpness-Aware Minimization (SAM) has emerged as a compelling alternative to stochastic gradient descent (SGD) as an optimizer for neural networks (Bahri et al., 2021). The core motivation of SAM is to not just to minimize the original training objective, but to also find a minimum with small Hessian norm (Wen et al., 2022). This motivation has been inspired by conventional wisdom that "flat" minima may correspond to better in-distribution generalization in comparison to sharper minima (Keskar et al., 2016; Jiang et al., 2019) (Foret et al., 2020). However, there has been a long-standing debate on whether this flatness is even a reliable predictor of generalization (Dinh et al., 2017; Kaddour et al., 2022; Andriushchenko et al., 2023b; Wen et al., 2023; Kaur et al., 2023). To this end, alternative theories about SAM have been put forth to explain its performance gains, typically by studying specific inductive biases of SAM in simple theoretical settings (Andriushchenko & Flammarion, 2022; Andriushchenko et al., 2023a; Behdin & Mazumder, 2023).

In this paper, we provide a complementary perspective to the discussion relating flatness to SAM's in-distribution gains. We argue that SAM can lead to certain *out-of-distribution* gains via a *feature diversifying effect*. Specifically, we consider datasets with multiple *redundant* features (e.g., an image of a cat may be recognized by its fur or eyes or tail, or of a truck by its wheels or its front or side doors). In such datasets, SGD is known to suffer from a "simplicity bias" towards learning only the simplest features that suffice for the task, over a range of datasets (Shah et al., 2020). However, we show that SAM overcomes this bias to learn multiple highly correlated features. Remarkably, this happens even though the SAM objective was not explicitly designed with this anti-simplicity, diversity bias. This bias makes SAM favorable in downstream tasks with distributed shifts where a diverse set of features become relevant for prediction.

Why does SAM lead to this feature-diversifying effect, even when learning only one of the features would suffice to minimize the objective? We construct a minimal toy setting with a simple and a complex feature to address this question. In the toy setting, SGD fails to capture an informative representation of the complex feature due to the simplicity bias. By contrast, SAM is able to learn a high-quality representation of both the simple and the complex feature. We argue that the key difference between the two algorithms can be seen directly in their respective gradient update steps. By decomposing the SAM gradient update step into two terms, we are able to identify two separate effects of SAM that lead to the feature-diversifying effect. The first effect is a balancing of the

weight of the training examples during each update step, leading to a more uniform update across the dataset. This effect is similar to other methods that explicitly re-weight examples to promote complex-feature learning such as GroupDRO (Sagawa et al., 2019) and Just Train Twice (Liu et al., 2021). Unlike these methods, SAM re-weights points implicitly through its update step. The second effect is a balancing of the effective step size taken to learn each feature: the effective learning rates of well-learned features are suppressed. These effects lead to balanced learning of all features, and thus a more diverse representation. SGD, by contrast, does not exhibit these balancing effects.

The feature diversity induced by SAM is particularly important in the setting in which the features which are redundant in the training distribution are not all necessarily predictive of the label in a downstream distribution. For example, the Waterbirds dataset (Sagawa et al., 2019) consists of images of water birds and land birds on backgrounds of water and land. In the training distribution, the background and bird type are highly correlated, but downstream, the background is not predictive of the bird type and vice versa. Ideally, we learn a representation amenable to adaptation to predict bird type, or background. We find that SAM indeed improves the performance of transfer learning with last-layer retraining on the downstream Waterbirds tasks, and on other similar datasets, include CelebA, CIFAR-MNIST, and DomainBed.

Our perspective is complementary to the large body of existing literature that discusses the in-distribution improvements of SAM and variants (Foret et al., 2020; Na et al., 2022; Zhang et al., 2023; Zhao et al., 2022; Rangwani et al., 2022; Wang et al., 2022; Meng et al., 2023). Importantly, this finding allows us to contextualize the small set of prior work that has discovered that SAM helps out-of-distribution performance (Wang et al., 2023; Cha et al., 2021; Bahri et al., 2021). We hope that this work will inspire future work to further understand the mechanisms of SAM and will motivate new algorithms for improving downstream performance on out-of-distribution data in similar contexts.

In summary, our contributions are as follows:

1. We identify a feature diversifying effect of SAM in settings in which data has multiple redundant features, and verify that this effect improves feature representation quality.

2. We construct and analyze a minimal toy setting consisting of multiple redundant features where SGD fails to capture an informative representation of all features, but SAM succeeds.

3. We demonstrate that the feature diversifying effect of SAM arises from two interpretable balancing effects: a balancing effect that reweights the training examples to be more uniform and a balancing effect that suppresses the effective learning rate of well-learned features.

4. We verify that the improved feature representations can be used to improve performance on out-of-distribution downstream tasks in realistic settings.

## 2 RELATED WORK

**Understanding Sharpness-Aware Minimization.** The success of SAM is commonly explained by the connection between the flatness of the landscape and improved generalization (see Section 5 of Foret et al. (2020)). The connection continues to inspire flatness-optimizing objectives for a wide range of tasks (Abbas et al., 2022; Na et al., 2022; Wang et al., 2023; Zhang et al., 2023; Zhao et al., 2022; Rangwani et al., 2022; Wang et al., 2022; Meng et al., 2023).

However, recent work has questioned whether sharpness and generalization are as linked as previously thought. On the one hand, it appears that sharper minima can generalize well after all, and on the other, even flat minima may sometimes generalize poorly (Dinh et al., 2017; Kaddour et al., 2022; Andriushchenko et al., 2023b; Wen et al., 2023; Kaur et al., 2023). In this backdrop, some works on understanding SAM have departed from flatness. These explicitly characterize the implicit bias of SAM in simple theoretical settings to show that it can improve in-distribution generalization (Andriushchenko & Flammarion, 2022; Behdin & Mazumder, 2023). Our work takes a different approach departing from both flatness and deriving the implicit regularization by looking at how the SAM perturbation changes each gradient step along the trajectory. Further, while these works focus on in-distribution generalization, we focus on the feature-diversifying effect of SAM in out-of-distribution and transfer learning settings.

**Benefits of SAM beyond in-distribution generalization.** SAM has been shown to improve performance in across various other settings such as meta learning (Abbas et al., 2022), domain generalization (Wang et al., 2023; Cha et al., 2021), label noise (Foret et al., 2020), transfer learning

in language models (Bahri et al., 2021). Our work augments these findings by offering one unifying factor that can explain SAM's gains beyond in-distribution generalization—SAM learns higher quality representations of hard-to-learn features.

**Feature learning in the presence of multiple predictive features.** It has been shown that neural networks learn some features more easily than others, a tendency that is dubbed as a "simplicity bias" (Shah et al., 2020; Kalimeris et al., 2019; Morwani et al., 2023; Rahaman et al., 2019). Besides different learning speeds, the presence of one feature also impacts the *quality* (i.e. probing error) of another feature in the learned representation as shown in (Pezeshki et al., 2021). This has inspired various fixes such as regularizers (Pezeshki et al., 2021), selecting freezing of parameters (Ye et al., 2023) and data augmentation (Plumb et al., 2021). This problem has also been addressed in a line of work aiming to improve worst-group error in the presence of spurious correlations, notably Sagawa et al. (2019), but we defer a detailed discussion to Appendix A. In contrast to these algorithms that are explicitly designed to address the simplicity bias, we show that SAM does so *without being explicitly incentivized to do so*.

We elaborate on further related work including the broader literature on shortcut learning and spurious correlations, feature diversity and finetuning, and variants of the SAM algorithm in Appendix A.

## 3 SETUP AND PRELIMINARIES

**Task.** We consider the setting of classification where we map input $x \in \mathcal{X}$ to output $y \in \mathcal{Y}$. Given training data $(x_1, y_1), (x_2, y_2), \ldots (x_n, y_n)$ where $x_i, y_i \sim D_{\text{train}}$, our goal is to learn a neural network classifier $f : \mathcal{X} \mapsto \mathcal{Y}$. We are interested in the diversity of the representations used by a classifier. Hence it is convenient to parameterize the neural network classifiers $f$ as linear classifiers on top of feature representations, i.e. $f(v, \theta; x) = \arg\max(v^\top \Phi_\theta(x))$, where $v \in \mathbb{R}^{k \times |\mathcal{Y}|}$ and $\Phi_\theta : \mathbb{R}^d \mapsto \mathbb{R}^k$. Note that the feature map $\Phi_\theta$ is itself a neural network. We use $w = (v, \theta)$ to denote the weights of the network when we do not need to discuss the features explicitly.

**Multiple predictive features.** In our setting, there are several predictive features. As an example, inspired by Sagawa et al. (2019), we consider the CelebA inputs as having two redundant features: a gender feature {*male, female*} and hair color {*blond, dark*}, both, in part, predictive of the label. Formally, each input $x$ has a set of features $a_1(x), a_2(x), \ldots a_m(x)$ where each $a_i(x) \in \mathcal{A}_i$ denotes the discrete (ground truth) value associated with the $i^{\text{th}}$ feature. We are particularly interested in the setting where several features are correlated with the label $y$ at varying strengths.

### 3.1 EVALUATING FEATURE QUALITY

In this work, we seek to compare the *feature quality* of the representations learned by different methods. To measure feature quality, our idea is to train a linear probe on the representation to discriminate the corresponding feature, and measure its performance (Rosenfeld et al., 2022; Kirichenko et al., 2022). Importantly, the training must be done on a dataset only the desired feature is correlated with the label, so that probe only picks up only this feature from the representation.

Formally, we construct the linear probe dataset as follows. Consider a *balanced* distribution $D_{\text{bal}}$ where there is an equal number of points from each configuration of the features, i.e., there are $|\mathcal{A}_1| \times |\mathcal{A}_2| \times \ldots |\mathcal{A}_m|$ subpopulations that have equal probability masses in $D_{\text{bal}}$. Then, for any feature $i$, we define the *feature-probing error* for this representation in terms the error of a linear probe $u$ in predicting the true feature value $a_i(x)$ when trained on $D_{\text{bal}}$. Formally,

$$\text{ProErr}_i(\theta) \stackrel{\text{def}}{=} \mathbb{E}_{x \sim D_{\text{bal}}}[\ell_{\text{0-1}}(u^\top \phi_\theta(x), a_i(x))], \text{ for } u = \arg\min \hat{E}_{x \sim D_{\text{bal}}}[\ell(u^\top a_i(x))], \quad (1)$$

where $\hat{E}_{x \sim D_{\text{bal}}}$ corresponds to the empirical distribution over training data from $D_{\text{bal}}$ used to train the linear probe and $\ell$ is some suitable classification loss such as the logistic loss.

### 3.2 TRAINING ALGORITHMS

**Empirical risk minimization via Stochastic Gradient Descent (SGD).** Stochastic gradient descent (SGD) is the *de facto* approach to minimizing the empirical risk over training data. Given a batch of training samples, $\{(x_1, y_1), \ldots (x_B, y_B)\}$, a loss function $\ell$, and model $w = (V, \theta)$, we define the empirical batch loss $\hat{\mathcal{L}}(w) \stackrel{\text{def}}{=} (1/B) \sum_{i=1}^{B} \ell(f(w; x_i), y_i)$. The SGD update is:

$$w \leftarrow w - \eta \nabla_w \hat{\mathcal{L}}(w), \quad (2)$$

Table 1: Comparing the in-distribution testing error and balanced distribution probing error for each feature for SAM and SGD, as defined in Section 3. For CelebA, the hard feature is gender, for Waterbirds, it is background, for CIFAR-MNIST, it is CIFAR, and for FMNIST-MNIST, it is FMNIST.

|  | Test | Easy ft. probe | Hard ft. probe | Test | Easy ft. probe | Hard ft. probe |
|---|---|---|---|---|---|---|
|  | | CelebA | | | Waterbirds | |
| SGD | $4.7 \pm 0.07$ | $10.9 \pm 1.15$ | $20.9 \pm 0.93$ | $5.2 \pm 0.13$ | $11.7 \pm 2.28$ | $21.8 \pm 0.99$ |
| SAM | $\mathbf{4.3 \pm 0.10}$ | $\mathbf{8.7 \pm 0.51}$ | $\mathbf{15.1 \pm 1.10}$ | $\mathbf{4.6 \pm 0.12}$ | $\mathbf{7.8 \pm 0.50}$ | $\mathbf{19.7 \pm 2.00}$ |
|  | | CIFAR-MNIST | | | FMNIST-MNIST | |
| SGD | $0.1 \pm 0.02$ | $0.1 \pm 0.02$ | $12.7 \pm 1.05$ | $0.5 \pm 0.52$ | $5.8 \pm 4.15$ | $11.6 \pm 1.93$ |
| SAM | $0.0 \pm 0.01$ | $0.1 \pm 0.03$ | $\mathbf{10.2 \pm 0.51}$ | $\mathbf{0.0 \pm 0.02}$ | $\mathbf{0.3 \pm 0.10}$ | $\mathbf{10.5 \pm 0.36}$ |

where $\eta > 0$ is the learning rate. Unless otherwise mentioned, we use the cross-entropy loss for multi-class classification and logistic loss for binary classification.

**Sharpness-Aware Minimization (SAM).** In recent years, SAM has been proposed as an alternative to SGD. For model weights $w$, the SAM update is:

$$w \leftarrow w - \eta \sum_{i=1}^{B} \nabla_w \hat{\mathcal{L}}(w)\big|_{\tilde{w}}, \text{ where } \tilde{w} \overset{\text{def}}{=} w + \rho \nabla_w \hat{\mathcal{L}}(w)/\|\nabla_w \hat{\mathcal{L}}(w)\|_2, \tag{3}$$

for some perturbation radius $\rho > 0$. We refer the reader to Foret et al. (2020) for a derivation.

**The SAM phantom parameter.** Comparing SGD (Equation 2) and SAM (Equation 3), notice that SAM takes a descent step in the direction of the gradient computed at different point: $\tilde{w}$, which we call the *phantom parameter*. The phantom parameter is computed by taking a ascent step of fixed norm in the direction of the gradient at the current parameter $w$. This is an important characterization which we exploit in the rest of this paper. By studying how the phantom parameter $\tilde{w}$ relates to the real parameter $w$, we can study how SAM changes the learning trajectory directly.

## 4 SAM IMPROVES FEATURE DIVERSITY

In this section, we demonstrate that SAM can empirically improve the quality of multiple redundant features, even when SGD would fail as a result of the simplicity bias (Shah et al., 2020). This quality is particularly important when the features relevant to downstream performance are unknown at training time, and thus necessary to learn a high quality representation of all available features. We evaluate SAM on datasets in which multiple features have been labeled, where SGD tends to learn a higher quality representation of the easier feature. With multiple labeled features, we can evaluate the quality of the representation of each feature individually. We first describe our experimental setup, and then present our results in which we compare the feature quality of SAM and SGD.

### 4.1 DATASETS AND MODELS

**Datasets.** We use four datasets in our experiments each annotated by two features: CelebA (Liu et al., 2015), Waterbirds (Sagawa et al., 2019), CIFAR-MNIST (binary) (Shah et al., 2020), and FMNIST-MNIST (5-class) (Kirichenko et al., 2022). The simplicity bias of SGD suggests that the easier of the two features will be learned better. Thus, we refer to the feature which attains the lower probing error for SGD trained models as the "easy" feature and the other has the "hard" feature. We describe the datasets in detail in Appendix B.

**Training setup.** Following prior work (Kirichenko et al., 2022; Sagawa et al., 2019), we train an ImageNet-pretrained ResNet-18 on CelebA and Waterbirds that has been initialized with weights pretrained on ImageNet. For CIFAR-MNIST, we train a three-layer MLP from scratch. We apply standard data augmentation and weight decay (detailed in Appendix B), and we tune both algorithms over different learning rates, and for SAM, we tune over the $\rho$ parameter. We select the optimal hyperparameters based on the validation set. For CIFAR-MNIST and CelebA, we train for 20 epochs, and for Waterbirds we train for 100 epochs. We repeat each experiment four times using different random seeds, and report the mean and standard deviation of the errors.

## 4.2 COMPARING SAM AND SGD

Our goal is to quantify the quality of the features learned by SGD-trained models vs. SAM-trained models. To this end, in all our datasets, we evaluate both the easy and hard features each model in terms of their feature probing error. As described in Section 3, we compute the probing error by training a last-layer probe on a small dataset in which the feature of interest is uncorrelated with the other feature. Then, we report the accuracy of this probe on a holdout test dataset in which the feature of interest is again uncorrelated with the other feature. In accordance with our main hypothesis, we find that SAM consistently achieves lower probing error for both features on all our datasets, in comparison to SGD (Table 1). We have thus verified that SAM implicitly improves the quality of multiple redundant features.

This suggests that even though SAM tries to optimize for the loss (and its sharpness) on the training distribution—and it does so successfully—under the hood, it also somehow improves feature quality on multiple *different*, balanced distributions in which the correlation between features is broken and examples are labeled by a feature of interest. We are interested in understanding how this unexpected "under the hood" improvement of the feature quality arises in SAM.

## 5 UNDERSTANDING SAM FEATURE LEARNING IN A TOY SETUP

In this section, we design a *minimal*, controlled setup that helps us isolate two core mechanisms within SAM. In our setup, data has an easier and a harder feature that are both predictive of the label. SGD exhibits the simplicity bias and learns a poor quality representation of the harder feature. In contrast, we show that SAM learns a good representation of both features.

Figure 1: Illustration of the toy example. **(i)** The toy data distribution. We vary the complexity of the spiral component of the data by tightening the spiral. **(ii)** Decision boundary of classifiers trained with SGD and with LSAM along a 2D slice where the other feature.

### 5.1 A MINIMAL SETUP

**Feature distribution.** We consider a 4D setup $\mathcal{X} = \mathbb{R}^4$ where the first two coordinates correspond to a *linear feature* (the "easier" feature) and the second two coordinates correspond to a *spiral feature* (the "harder" feature). We denote the individual components $x = [x_{\mathsf{easy}}, x_{\mathsf{hard}}]$. The associated attributes (see Section 3) for each feature $a_1(x), a_2(x)$ are such that $a_1(x) = 1$ if $x$ resides on the right branch of the linear feature, and zero otherwise; similarly, $a_2(x) = 1$ if $x$ resides on the right spiral, and zero otherwise. We assume a binary label $y \in \{-1, 1\}$ such that each feature independently is fully predictive of it i.e., $y = a_1(x)$ and $y = a_2(x)$. We can vary the *complexity* of the spiral feature by varying the number of rotations of the spiral. Here, the complexity refers to the amount of rotation of the spiral, in degrees.

**Disentangled architecture.** We are interested in a setting where it is possible to *precisely* measure the quality of representations. To this end, we create a network with an explicitly disentangled representation space (rather than hoping that a trained linear probe would precisely disentangle it). Specifically, we define an architecture whose last-layer representation takes the form

$$\Phi_\theta(x) \overset{\mathsf{def}}{=} [\Phi_{\mathsf{easy}}(x), \Phi_{\mathsf{hard}}(x)] \overset{\mathsf{def}}{=} [\Phi_\theta([x_{\mathsf{easy}}, \mathbf{0}]), \Phi_\theta([\mathbf{0}, x_{\mathsf{hard}}])], \tag{4}$$

where $\Phi_\theta \colon \mathbb{R}^4$ to $\mathbb{R}$ is a three-layer neural network. This is followed by last layer weights $v = [v_{\mathsf{easy}}, v_{\mathsf{hard}}]$ such that the final classification takes the form, $\mathrm{sign}(f_w(x)) = \mathrm{sign}(v_{\mathsf{easy}}\Phi_{\mathsf{easy}}(x) + v_{\mathsf{hard}}\Phi_{\mathsf{hard}}(x))$. where $w = (v, \theta)$ are the parameters of the neural network. Then, we can think of our feature probe for either features as simply $\Phi_{\mathsf{easy}}$ and $\Phi_{\mathsf{hard}}$ respectively.

**Probing error.** Since we have a disentangled representation space, measuring the probing error is easy—there is no need to train a linear probe, since each feature is represented separately in the last layer. We can simply measure the error $\ell_{0\text{-}1}(\Phi_{\mathsf{easy}}(x), y)$ or $\ell_{0\text{-}1}(\Phi_{\mathsf{hard}}(x), y)$ to compute the probing error for the easy and hard feature, respectively.

**LSAM: A simplification of SAM.** For this controlled study, we consider a minimal version of SAM where we only perturb the last layer $V$; we call this LSAM. While the final descent of LSAM still updates all parameters; it is only the *phantom* ascent that is restricted to the last layer.

Specifically, following the Section 3 notation, the phantom parameters take the following simple form:

$$\tilde{v} \stackrel{\text{def}}{=} v + \rho\nabla_v\hat{\mathcal{L}}(v,\theta)/\|\nabla_v\hat{\mathcal{L}}(v,\theta)\|_2, \quad \tilde{\theta} \stackrel{\text{def}}{=} \theta. \tag{5}$$

This is nearly identical to the SAM phantom parameter described by Equation 3, except that only $v$ is perturbed, and not $\theta$. As we will shortly see (Section 6), the above simplified version of SAM is sufficient (and in fact, helpful) in cleanly identifying our two fundamental mechanisms that can be linked to improved feature quality.

### 5.2 LSAM LEARNS THE HARDER SPIRAL FEATURE, WHILE SGD DOES NOT

We note that in this dataset, both SAM and SGD learn the label $y$ perfectly. However, the behaviors are starkly different at the feature-level. In Figure 1, we plot the contour of $\Phi_{\text{easy}}$ and $\Phi_{\text{hard}}$ which correspond to how the models have learned either features. We observe, visually, that SGD has no trouble learning the linear feature but has a poor representation of the hard spiral feature. However, LSAM manages to learn a good representation of both features. We quantify this observation by measuring the spiral feature's probing error for varying complexities of the spiral feature (i.e., varying tightness levels), and for varying values of the SAM perturbation radius $\rho$ (Figure 2(A)). We find that while the learned spiral

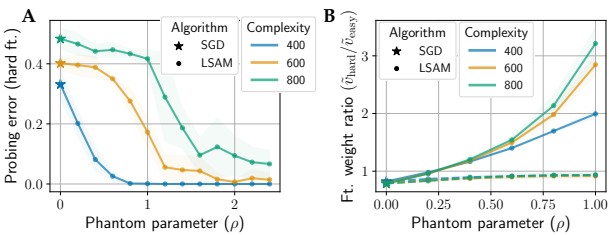

Figure 2: (**A**) Test-set probing error of harder feature as a function of the phantom parameter $\rho$, for multiple complexities for the hard feature. SGD corresponds with $\rho = 0$. (**B**) Phantom weight ratio as a function of $\rho$. We plot $\tilde{v}_{\text{hard}}/\tilde{v}_{\text{easy}}$ (solid lines) and $v_{\text{hard}}/v_{\text{easy}}$ (dashed lines) for different values of $\rho$ when running LSAM. Note that SGD corresponds to when $\rho = 0$. As $\rho$ increases, the ratio $\tilde{v}_{\text{hard}}/\tilde{v}_{\text{easy}}$ increases.

feature's quality is poorest when $\rho = 0$ (i.e., SGD), it progressively improves as we increase $\rho$. In effect, this toy example isolates a core property of SAM that sets it apart from SGD: SAM learns higher quality representations of hard-to-learn features compared to SGD, even in the presence of a fully predictive easy feature. This is analogous to the feature-diversifying effect we observe in real-world datasets in Section 4.

## 6 WHY DOES SAM LEARN THE HARD-TO-LEARN FEATURE?

We have so far established that LSAM achieves significantly lower probing error on a hard-to-learn spiral feature in comparison to SGD (See Figure 2(A)). In this section, we will describe the two mechanisms by which LSAM offers this feature-diversifying benefit without being explicitly trained to do so—we will argue that LSAM re-weights the training examples to increase weight on points that may be helpful for learning the hard feature, and that LSAM increases the effective step size of updates that fit the hard feature.

### 6.1 SAM RE-BALANCES PHANTOM PARAMETERS

The first step in understanding this mechanism is to understand how the phantom parameters $\tilde{v}_{\text{easy}}$ and $\tilde{v}_{\text{hard}}$ behave as we vary the perturbation radius $\rho$. We plot the ratio $\tilde{v}_{\text{hard}}/\tilde{v}_{\text{easy}}$ as a function of $\rho$ in Figure 2(B), which shows that the ratio increases as we increase $\rho$. This implies that as $\rho$ is increased, LSAM places more relative weight on the hard feature during the update step, as described by Equation 3. Note that the re-balancing effect is not the result of the real weight $v_{\text{hard}}$ and $v_{\text{easy}}$ changing, but rather the result of the perturbation applied by LSAM when computing the phantom parameters. The ratio of the real parameters $v_{\text{hard}}/v_{\text{easy}}$ remains nearly constant as a function of the perturbation radius $\rho$ (Figure 2(B), dashed line).

### 6.2 THE FEATURE-DIVERSIFYING EFFECTS OF SAM

With knowledge that LSAM reweights the ratio $\tilde{v}_{\text{hard}}/\tilde{v}_{\text{easy}}$, our main idea is to decompose the gradient update into two terms in which we can interpret the effect of this reweighting. For binary

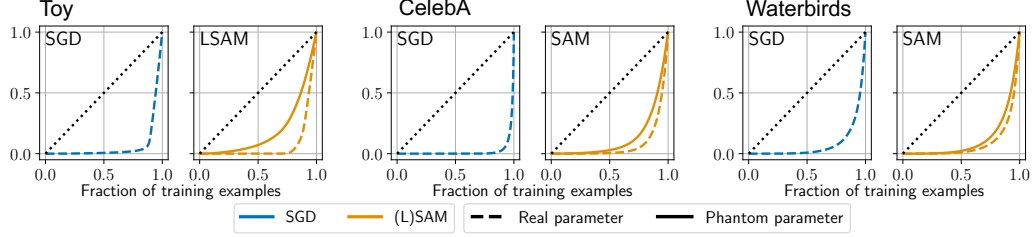

Figure 3: Lorenz curves for the real and phantom importance weight $\lambda_i$ and $\tilde{\lambda}_i$. The dotted diagonal line represents the Lorenz curve for a uniform distribution. The closer this curve is to this diagonal, the more equally the importance weights are spread. In blue, we plot the Lorenz curves for an SGD checkpoint. In orange, we plot the Lorenz curves for an LSAM checkpoint. The update step gradient is computed at real parameter for SGD, and the phantom parameter for SAM. We include the curves for the toy (left), CelebA (center), and Waterbirds (right).

classification, logistic loss on sample $(x_i, y_i)$, we can write the gradient update to parameters $\theta$ as:

$$\nabla_\theta \hat{\mathcal{L}} = \underbrace{\sigma\big(-y_i(v_{\mathsf{easy}}\Phi_{\mathsf{easy}}(x_i) + v_{\mathsf{hard}}\Phi_{\mathsf{hard}}(x_i))\big)}_{\text{Importance weighting } \lambda_i} \cdot \underbrace{\big(v_{\mathsf{easy}}\nabla_\theta\Phi_{\mathsf{easy}}(x_i) + v_{\mathsf{hard}}\nabla_\theta\Phi_{\mathsf{hard}}(x_i)\big)}_{\text{Sum of feature gradients } g_i},$$

(6)

where $\sigma$ is the sigmoid function. If we replace $v_{\mathsf{hard}}$ and $v_{\mathsf{easy}}$ with the phantom parameters $\tilde{v}_{\mathsf{hard}}, \tilde{v}_{\mathsf{easy}}$, we get analogous terms $\tilde{\lambda}_i$ and $\tilde{g}_i$ for LSAM. The scalar *importance weighting* term $\lambda_i$ represents the contribution of each individual example to the final loss gradient. The vector *sum of feature gradients* term $g_i$ is a weighted combination of the individual feature gradients. The effect of LSAM on the importance weighting term is example-specific: the degree to which each point is up-weighted by LSAM $\tilde{\lambda}_i/\lambda_i$ depends on the example. However, the effect of LSAM on the sum of feature gradients term is constant across all examples in a batch: the degree to which the effective learning rate of each feature is up-weighted by LSAM, $\tilde{v}_{\mathsf{hard}}/v_{\mathsf{hard}}$ and $\tilde{v}_{\mathsf{easy}}/v_{\mathsf{easy}}$, is the same for all examples within a batch.

In the remainder of the section, we explore two effects that arise when $\tilde{v}_{\mathsf{hard}}/\tilde{v}_{\mathsf{easy}} \gg v_{\mathsf{hard}}/v_{\mathsf{easy}}$ that enable LSAM to learn the harder feature.

**Effect one: the importance weighting effect.** We consider the effect of increasing $\tilde{v}_{\mathsf{hard}}/\tilde{v}_{\mathsf{easy}}$ on the $\tilde{\lambda}_i$ term in Equation 6, which corresponds to how heavily each point is weighted in the gradient. At a high level, we argue that this effect leads to two key beneficial behaviors that can be tested experimentally: first, examples are weighted more uniformly, and second, examples that are poorly fit by the hard feature are upweighted, even when they are well-fit by the easy feature. Both of these effects promote the learning of the harder feature by encouraging additional points to contribute to the learning of the harder feature that would otherwise have contributed negligibly. In particular, the second effect leads to the up-weighting of points that are incorrectly fit by the harder feature but have a low loss because they are fit well by the easy feature. The effect of LSAM is similar to the intervention imposed by algorithms that explicitly up-weight examples that belong to the minority class in imbalanced datasets (Liu et al., 2021; Sagawa et al., 2019), but it is remarkable that LSAM induces this importance weighting effect *implicitly* without being designed to do so.

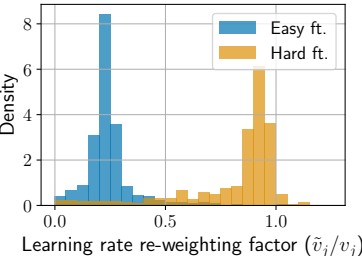

Figure 4: The learning rate re-weighting factor for the easy feature $\tilde{v}_{\mathsf{easy}}/v_{\mathsf{easy}}$ and the hard feature $\tilde{v}_{\mathsf{hard}}/v_{\mathsf{hard}}$, plotted as a distribution over batches in the dataset (see Equation 6).

To verify that points are more uniformly weighted in the update step, we visualize the distribution of their importance weights as a Lorenz curve, which plots the cumulative fraction of the total importance contributed by the top $k$ points as a function of $k$ (Figure 3). This means that the closer the Lorenz curve is to a diagonal line, the more uniformly the importance weights are distributed across all points. We use the Lorenz curves to compare the behavior of an LSAM step to an SGD

step given fixed model parameters (Figure 3, left, comparing dashed to solid). We also compare the Lorenz curves of the importance weights computed using the real parameters and computed using the phantom parameters for a model that has been trained with SGD and one that has been trained with LSAM (Figure 3, left, comparing solid line in LSAM plot to dashed line in SGD plot). We find that the importance weights are more uniformly weighted under LSAM in both settings. We plot additional Lorenz curves corresponding to different points during training in Appendix G.

To verify that points that are well fit by the easy feature but poorly fit by the hard feature are upweighted. We need a metric to measure the the degree to which the easy and hard features are fit by the model. In our toy setting, we can compute the contribution of each feature explicitly. We can thus measure the degree to which the easy and hard features are fit by the model as their contribution signed by the label: $y\Phi_{\text{easy}}(x)$ and $y\Phi_{\text{hard}}(x)$. When either term is positive, the point is classified correctly by the corresponding feature, with the magnitude of the term indicating the margin. In order to summarize how points are weighted as a function of the signed contributions, we partition the points into bins based on $y\Phi_{\text{easy}}(x)$ and $y\Phi_{\text{hard}}(x)$. Thus, we plot the median importance weight for each bin, which gives us the relationship between the importance weight of each example and their signed feature contribution. We see that typical examples that are poorly-fit by the easy feature ($y\Phi_{\text{easy}}(x)$ is small) are assigned less importance by SAM, and examples that are poorly-fit by the hard feature ($y\Phi_{\text{hard}}(x)$ is small) are assigned more importance by SAM (Figure 5). This verifies our claim that LSAM up-weights points that are well-fit by the easy feature but poorly-fit by the hard feature in the toy setting. We plot additional importance weight plots corresponding to different points during training in Appendix G.

**Verifying the importance weighting effect in a real-world setup.** We have thus far shown that LSAM induces the importance weighting effect in the toy setting. However, what happens when we relax our assumptions and move to a more realistic setting? We aim to test the

Figure 5: Median importance weighting as a function of the contribution of each feature. We partition the data into bins based on the contribution of the easy and hard features ($yv_{\text{easy}}\Phi_{\text{easy}}$ and $yv_{\text{easy}}\Phi_{\text{hard}}$), as defined in Section 6.2. For each of these bins, we plot the median importance weight term $\lambda_i$ for the points in the bin. We include the corresponding plots for the toy (top), CelebA (center), and Waterbirds (bottom).

importance weighting effect when we run the usual SAM algorithm without a disentangled representation space on real-world datasets. We can evaluate importance weight uniformity directly. We plot the Lorenz curve for the importance weights evaluated at both the real parameter and the phantom parameter for CelebA and Waterbirds in Figure 3. We see that the points are more uniformly weighted under a SAM perturbation, suggesting that the importance weighting effect is present in the real-world datasets as well.

To verify that LSAM up-weights points that are well-fit by the easy feature but poorly-fit by the hard feature, we used the disentangled representation space to measure the degree to which the easy and hard features are fit by the model. However, in the real setting, we cannot directly measure the contribution of each feature. Instead, we rely on extracting the feature contributions from the linear probes $v_{\text{easy}}$ and $v_{\text{hard}}$ that best classifies the respective feature (refer to Section 3). With these probes, we can define the signed contribution of the easy and hard feature as $yv_{\text{easy}}^\top\Phi(x)$ and $yv_{\text{hard}}^\top\Phi(x)$, respectively. Thus, we plot the median importance weight for both features as a function of $yv_{\text{easy}}^\top\Phi(x)$ and $yv_{\text{hard}}^\top\Phi(x)$ for CelebA and Waterbirds. Analogous to the toy experiment, we see that examples that are poorly-fit by the easy feature are assigned less importance by SAM, and examples that are poorly-fit by the hard feature are assigned more importance by SAM (Figure 5).

**Effect two: the learning rate scaling effect.** We now turn our attention to the second term, $\tilde{g}_i$, from Equation 6. The $\tilde{g}_i$ term is a weighted combination of the feature gradients with weights $\tilde{v}_{\text{easy}}$ and $\tilde{v}_{\text{hard}}$ corresponding to the easy and hard feature respectively. These weights can be viewed as the "learning rates" as we take gradients steps in the direction of learning different fea-

tures. Notably, these learning rates are constant across points within a batch. When LSAM causes $\tilde{v}_{\text{hard}}/\tilde{v}_{\text{easy}} \gg v_{\text{hard}}/v_{\text{easy}}$, $\tilde{g}_i$ corresponds with a larger effective step size associated with the hard feature in comparison to $g_i$. As a result the effective learning rate for the hard feature is higher and leads to more progress on the hard feature. We plot the ratio between the learning rates of the phantom parameters and the real parameters $\tilde{v}_{\text{hard}}/v_{\text{hard}}$ and $\tilde{v}_{\text{easy}}/v_{\text{easy}}$ for each feature in Figure 4. We see that LSAM increases the learning rate for hard features and decreases the learning rate for each feature. We call this the *learning rate scaling effect*. Unlike the importance weighting effect which acts on each data point separately, the learning rate scaling effect is more global across the entire training batch.

Unfortunately, we cannot directly verify the learning rate scaling effect without a disentangled representation space. We discuss this further in Appendix C.1.

**Contribution of each effect.** To recap, we show that LSAM induces (i) an importance weighting effect and (ii) a learning rate scaling effect, both of which in turn arise from by rebalancing the phantom parameters. We have verified that each of these effects are present in the toy example, and we would intuitively expect each effect to have a feature-diversifying effect. But how do these effects compare in improvement in feature quality, and which effect if any is more important? Are these effects simply correlations caused by SAM's improved features, or do these effects causally improve SAM?

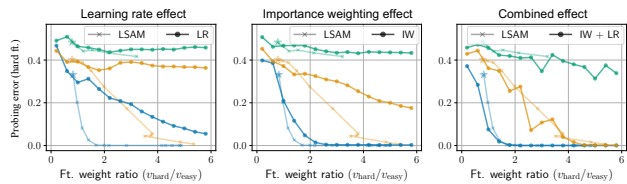

Figure 6: Simulations of the learning rate, importance weighting, and combined effects. Both effects offer gains in hard feature probing accuracy and together account for all the benefits offered by LSAM in the toy setting. See Appendix C for a detailed description of the simulations.

We answer these questions in the toy setting, where it is possible to explicitly add or remove each effect. To isolate each effect, we train with an algorithm that applies only one effect at a time: either the importance weighting effect or the learning rate effect.

To isolate the importance weighting effect, we train with an algorithm in which we fix the ratio $v_{\text{hard}}/v_{\text{easy}} = v^*_{\text{hard}}/v^*_{\text{easy}}$ to a constant value in the importance weighting,

$$\lambda_i := \sigma\big(-y_i(v^*_{\text{easy}}\Phi_{\text{easy}}(x_i) + v^*_{\text{hard}}\Phi_{\text{hard}}(x_i)\big) \qquad g_i \text{ is as usual for SGD} \qquad (7)$$

where $v^*_{\text{easy}} = 1$ and $v^*_{\text{hard}}$ is a hyperparameter.

We similarly isolate the learning rate effect and test the combination of both effects by fixing $v_{\text{hard}}/v_{\text{easy}}$ analogously. The details are described precisely in Appendix C.

In order to compare these algorithms with LSAM, we compute the probing error for the hard feature against the ratio between the hard feature contribution and the easy feature contribution. For our algorithms, we have manually set this value $v^*_{\text{hard}}/v^*_{\text{easy}}$ (see Equations 7). For LSAM, we consider this value to be the mean value of $v_{\text{hard}}/v_{\text{easy}}$ over training (From Figure 2(B)). This leaves both our new algorithms and LSAM to be directly comparable as a function of this ratio, which we plot (Figure 6). We see that both individual interventions (Figure 6, left and center) improve the probing error for the hard feature. This suggests that both effects are causal and also significant (offer comparable gains) in explaining LSAM's improved spiral probing error in the toy setting. Further, the combined effect (Figure 6, right) matches the performance of LSAM and thus appears to provide a complete picture of understanding LSAM in our toy.

## 7 CONCLUSION

In this work, we offer insight into the dynamics of SAM that lead to the improvement of feature diversity, in contrast with the usual sharpness-based arguments. We have shown how SAM can promote balanced feature learning in the presence of multiple redundant features, and that this can lead to improved performance on out-of-distribution tasks. We have also demonstrated that SAM leads to improvements in feature quality for transfer learning on real data, including the Waterbirds dataset, CelebA, CIFAR-MNIST, and DomainBed. We hope that our insights provide a new perspective on the dynamics of SAM without relying on the flatness-based arguments, and that this work will inspire future work to further understand the mechanisms of SAM and will foster new algorithms for improving downstream performance on out-of-distribution data in similar contexts.

## ACKNOWLEDGEMENTS

We would like to thank Christina Baek, Jeremy Cohen, and Suhas Kotha for their feedback.

This material is based upon work supported by the National Science Foundation Graduate Research Fellowship under Grant No. DGE2140739. Any opinion, findings, and conclusions or recommendations expressed in this material are those of the authors(s) and do not necessarily reflect the views of the National Science Foundation. This work was supported in part by the AI2050 program at Schmidt Sciences (Grant #G2264481). We gratefully acknowledge the support of Apple.

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

## A ADDITIONAL RELATED WORK

**Shortcut learning and spurious correlations.** Many works have demonstrated shortcut learning in neural networks image and natural language classification tasks (Geirhos et al., 2018; 2020; Scimeca et al., 2021; Yang et al., 2022; Hermann et al., 2020; Hermann & Lampinen, 2020; Brendel & Bethge, 2019; Alcorn et al., 2019; Shetty et al., 2019; Singla & Feizi, 2021; Rosenfeld et al., 2018; Kolesnikov & Lampert, 2016; Moayeri et al., 2022; Gururangan et al., 2018; McCoy et al., 2019; Xiao et al., 2020; Sagawa et al., 2019). In contrast, we focus purely on feature *probing* error which looks at how well a feature is captured in the representation irrespective of how much the final classifier relies on different features. We also care about probing error with respect to all predictive features and do not delineate specific features as spurious. Recent works (Kirichenko et al., 2022; Izmailov et al., 2022; Rosenfeld et al., 2022) have shown that just last layer retraining on a suited dataset can significantly overcome reliance on spurious correlations suggesting that the non-spurious features are reasonably well-learnt by the representation.

**Feature diversity and finetuning.** Multiple methods have been proposed to improve feature diversity to evade the simplicity bias (Teney et al., 2022), ignore spurious features (Asgari et al., 2022), improve transfer learning (Pagliardini et al., 2022), and improve generalization (Jain et al., 2023). Fine-tuning and other retraining has been a popular method to adopt learned features to tasks other than the pre-training objective (Pan & Yang, 2009). Retraining has been adopted to recover from failure modes (Rosenfeld et al., 2022; Kirichenko et al., 2022; Kumar et al., 2022) and to improve

generalization on novel tasks (Radford et al., 2021; Sharif Razavian et al., 2014; Huh et al., 2016; Sun et al., 2017; Mahajan et al., 2018; Kolesnikov et al., 2020; Zhai et al., 2019; Dosovitskiy et al., 2020; Neyshabur et al., 2020; Abnar et al., 2021; Kornblith et al., 2019).

**Alternative SAM variants.** Multiple variants have been proposed to generalization notions of flatness (Kwon et al., 2021; Zhuang et al., 2022), to improve the efficiency of SAM (Du et al., 2022; 2021; Liu et al., 2022), and to generalize SAM to different optimizers (Sun et al., 2023).

## B OMITTED EXPERIMENTAL DETAILS

**Description of datasets.** CelebA is a large-scale face attributes dataset with 40 binary attributes. Following Kirichenko et al. (2022), we train a classifier to predict the "hair color" attribute, with values {blond, dark-hair}. However, the "gender" attribute, with values {female, male} is highly correlated with "hair color" due to the imbalance in the dataset, and thus can additionally be useful to predict the label, for the training distribution. Waterbirds is a synthetic dataset consisting of images of birds superimposed onto different backgrounds. The data is annotated by "bird type", with values {water bird, land bird}, which refers to whether the pictured bird primarily lives in water or on land. The "background" attribute, with values {water, land}, is highly correlated with "bird type". CIFAR-MNIST is a dataset consisting images from MNIST (LeCun et al., 1998) and CIFAR-10 (Krizhevsky et al., 2009) that have been concanetated together, where the label of the CIFAR image and the label of the MNIST image are perfectly correlated for all training examples. Following Shah et al. (2020), we restrict the dataset to the binary setting. The "CIFAR" attribute can attain values {airplane, automobile} and refers to the label of the CIFAR component of the image. Similarly, the "MNIST" attribute can attain values {0, 1}. For all datasets, we use the standard train/validation/test split, and when a validation set is not provided, we use a random 90/10 split of the training set. For computational efficiency, we down-scale the CelebA images to $64 \times 64$ pixels, and the Waterbirds images to $96 \times 96$ pixels. For FashionMNIST-MNIST dataset (Xiao et al., 2017), we restrict to the first five classes, associated with the digits 0–5 of MNIST.

**Reporting test error.** Since the validation and testing datasets of Waterbirds and CelebA differ in distribution from the training set, to be consistent with Kirichenko et al. (2022) when reporting the testing error, we weight the testing error of each group by its corresponding frequency in the training dataset.

**Architectures** For the experiments involving CIFAR-MNIST and variants, we train on a simple convolutional architecture including three convolutional layers, followed by ReLUs, with a final linear decoding layer. The architecture is defined by the following pseudo-PyTorch:

```
torch.nn.Sequential(
    torch.nn.Conv2d(3, 32, kernel_size=5, stride=2, padding=2),
    torch.nn.ReLU(inplace=True),
    torch.nn.Conv2d(32, 64, kernel_size=3, stride=2, padding=1),
    torch.nn.ReLU(inplace=True),
    torch.nn.Conv2d(64, 128, kernel_size=3, stride=2, padding=1),
    torch.nn.ReLU(inplace=True),
    torch.nn.Flatten(),
    torch.nn.Linear(n_features, num_classes)
)
```

For the experiments involving CelebA and Waterbirds, we use an ImageNet-pretrained ResNet-18 as specified by He et al. (2016).

For the toy experiments, the representation neural network $\Phi_\theta$ had the architecture specified by the following pseudo-PyTorch:

```
torch.nn.Sequential(
    torch.nn.Linear(n_features, 100),
    torch.nn.LayerNorm(100),
    torch.nn.ReLU(),
    torch.nn.Linear(100, 100),
    torch.nn.LayerNorm(100),
    torch.nn.ReLU(),
```

```
        torch.nn.Linear(100, n_classes)
)
```

**Parameters and sweeps.**   For the toy experiments, we choose a constant learning rate of $0.01$, a batch size of 5, 300 training points, no momentum, and no weight decay.

For the CIFAR-MNIST and FMNIST-MNIST experiments, we sweep over the learning rates $\{0.01, 0.05, 0.1\}$ and the phantom hyperparameter $\rho$ over $\{0.0, 0.01, 0.03, 0.05, 0.07, 0.1, 0.2\}$. We use a batch size of 100, a cosine learning rate schedule, a momentum parameter of 0.9, and no weight decay. We normalize the images by the mean pixel value. Otherwise, we do not use data augmentation.

For the CelebA and Waterbirds experiments, we sweep over the learning rates $\{0.0005, 0.001, 0.005, 0.01\}$ and the $\rho$ parameter $\{0.0, 0.01, 0.02, 0.05, 0.07\}$. We use a batch size of 128, a cosine learning rate schedule, a momentum parameter of 0.9, and a weight decay of $10^{-4}$. We use data augmentation described by the following pseudo-PyTorch:

```
WATERBIRDS_TRANSFORMS_AUGMENT = transforms.Compose([
    transforms.RandomResizedCrop(96,
        scale=(0.7, 1.0),
        ratio=(0.75, 4./3.),
        interpolation=InterpolationMode.BILINEAR),
    transforms.RandomHorizontalFlip(),
    transforms.ToTensor(),
    transforms.Normalize(WATERBIRDS_MEAN, WATERBIRDS_STD)
])

CELEBA_TRANSFORM = transforms.Compose([
    transforms.Resize((64, 64)),
    transforms.ToTensor(),
    transforms.Normalize(CELEBA_MEAN, CELEBA_STD)
])
```

## C  ADDITIONAL DETAILS FOR THE TOY SETTING

**Algorithms for the interventions**   To simulate only the importance weighting effect, we re-weight points in the importance weighting term $\lambda_i$ to the weight of each point computed as if we had fixed the feature ratio. Similarly, to simulate the learning-rate effect, we re-weight each feature in the sum of feature gradients term $g_i$ by the weight that would assigned if we had fixed the feature ratio. In order to verify that each effect is important and to quantify the relative strenght of each effect, we intervene individually on each effect, separately, by fixing the ratio $v_{\mathsf{easy}}/v_{\mathsf{hard}}$, *but only when computing the term corresponding with that effect*. This means that the loss gradient for the importance weighting intervention is computed as,

$$g_{\mathsf{i.w.}}(\theta) = \sum_{i=1}^{B} \sigma_{y_i} \left( v_{\mathsf{easy}}^{*} \Phi_{\mathsf{easy}}(x_i) + v_{\mathsf{hard}}^{*} \Phi_{\mathsf{hard}}(x_i) \right) \left( v_{\mathsf{easy}} \nabla_\theta \Phi_{\mathsf{easy}}(x_i) + v_{\mathsf{hard}} \nabla_\theta \Phi_{\mathsf{hard}}(x_i) \right),$$
(8)

and the loss gradient for feature gradient effective learning rate intervention is computed as,

$$g_{\mathsf{l.r.}}(\theta) = \sum_{i=1}^{B} \sigma_{y_i} \left( v_{\mathsf{easy}} \Phi_{\mathsf{easy}}(x_i) + v_{\mathsf{hard}} \Phi_{\mathsf{hard}}(x_i) \right) \left( v_{\mathsf{easy}}^{*} \nabla_\theta \Phi_{\mathsf{easy}}(x_i) + v_{\mathsf{hard}}^{*} \nabla_\theta \Phi_{\mathsf{hard}}(x_i) \right).$$
(9)

For both equations, $v_{\mathsf{easy}}^{*}$ and $v_{\mathsf{hard}}^{*}$ are fixed and do not change during training. We train with SGD, but we compute the gradient as described in Equations 8 and 9.

To simulate the effects together, we combine the two interventions by fixing the ratio $v_{\mathsf{easy}}/v_{\mathsf{hard}}$ in both terms. This means that the loss gradient for the combined intervention is computed as,

$$g_{\mathsf{c.}}(\theta) = \sum_{i=1}^{B} \sigma_{y_i} \left( v_{\mathsf{easy}}^{*} \Phi_{\mathsf{easy}}(x_i) + v_{\mathsf{hard}}^{*} \Phi_{\mathsf{hard}}(x_i) \right) \left( v_{\mathsf{easy}}^{*} \nabla_\theta \Phi_{\mathsf{easy}}(x_i) + v_{\mathsf{hard}}^{*} \nabla_\theta \Phi_{\mathsf{hard}}(x_i) \right).$$
(10)

## C.1 THE LEARNING RATE EFFECT IS DIFFICULT TO VERIFY IN PRACTICE.

Unfortunately, in the absence of explicitly disentangled architecture (like in our toy setting), it is difficult to estimate the learning rate scaling effect. The key challenge in achieving a similar observation of the learning-rate effect on real data is understanding precisely how features are represented. Since, SAM perturbs the weights at every layer (unlike LSAM), we would need to understand how each feature is represented at every layer of the neural network. Based on the general principle that SAM perturbs weights as a function of how much they affect the output, we suspect, intuitively, that well-learned features will be inhibited by SAM at every layer. While we cannot verify the effect explicitly for a realistic setup, we believe it contributes to SAM's gains in feature probing error.

## D   THEORETICAL INTUITION AND ANALYSIS

In this section, we present a brief discussion of the theoretical intuition behind our results. In particular, we will aim to understand the importance weighting effect.

**Preliminaries.**   For simplicity, we will aim to understand the dynamics when training with a batch is a single example $x$. We will assume, as is typical, that we have a neural network,

$$f_x(\theta) = w^\top \phi_\theta(x) \tag{11}$$

parameterized by $\theta$, and where $\phi_\theta(x)$ is the representation of $x$ under the neural network. We assume that the neural network is differentiable and well-approximated by a first-order Taylor expansion. We assume that the loss function $\mathcal{L}(\theta) = \exp(-yf(\theta))$ is exponential, though note that cross-entropy is almost identical to exponential, for correctly classified points. Finally, recall from the main body of the paper that we can separate the loss gradient into two terms,

$$\nabla_\theta \mathcal{L}(\theta) = -y \underbrace{\exp(-yf(\theta))}_{\text{importance weighting term } \lambda} \underbrace{\nabla_\theta f(\theta)}_{\text{feature gradient term } g}. \tag{12}$$

We define the phantom importance weighting parameter $\tilde{\lambda}$ as the importance weighting term evaluated at the phantom parameter $\tilde{\theta}$.

**Phantom parameter.**   Recall that the SAM perturbation with exponential loss is defined as,

$$\tilde{\theta} = \theta + \rho \frac{\nabla_\theta \mathcal{L}(\theta)}{\|\nabla_\theta \mathcal{L}(\theta)\|} \tag{13}$$

$$= \theta + \rho \frac{\nabla_\theta \exp(-yf_x(\theta))}{\|\nabla_\theta \exp(-yf_x(\theta))\|} \tag{14}$$

$$= \theta - \rho \frac{y \exp(-yf_x(\theta)) \nabla_\theta f_x(\theta)}{\|\exp(-yf_x(\theta)) \nabla_\theta f_x(\theta)\|} \tag{15}$$

$$= \theta - \rho y \frac{\nabla_\theta f_x(\theta)}{\|\nabla_\theta f_x(\theta)\|} \tag{16}$$

**Importance weighting.**   We first aim to understand the ratio between the phantom and real importance weighting term $\tilde{\lambda}/\lambda$. We have,

$$\tilde{\lambda}/\lambda = \exp\left(-yf(\tilde{\theta})\right) / \exp\left(-yf(\theta)\right) \tag{17}$$

$$= \exp\left(-y(f(\tilde{\theta}) - f(\theta))\right) \tag{18}$$

which arises from plugging in the definition of the importance weighting term with exponential loss. For convenience, we will consider the log of the ratio,

$$\log\left(\tilde{\lambda}/\lambda\right) = -y(f(\tilde{\theta}) - f(\theta)). \tag{19}$$

The effect is based upon a first-order Taylor expansion of the logit function $f(\theta)$,

$$f(\tilde{\theta}) = f(\theta) - \rho\, y\, \nabla_\theta f(\theta)^\top \frac{\nabla_\theta f(\theta)}{\|\nabla_\theta f(\theta)\|} + \mathcal{O}\left(\rho^2\right) \tag{20}$$

$$\approx f(\theta) - \rho\, y\, \|\nabla_\theta f(\theta)\|. \tag{21}$$

We can plug this into the expression for the ratio of importance weighting terms, and we get,

$$\log\left(\tilde{\lambda}/\lambda\right) \approx -y\left(f(\theta) - \rho\, y\, \|\nabla_\theta f(\theta)\| - f(\theta)\right) \tag{22}$$

$$= \rho\, \|\nabla_\theta f(\theta)\| \tag{23}$$

$$\tag{24}$$

since $y^2 = 1$. We can see that the ratio of importance weighting terms is approximately proportional to the magnitude of the feature gradient $\|\nabla_\theta f(\theta)\|$.

For a better sense of the dynamics, we compute $\|\nabla_\theta f(\theta)\|$ for a few different architectures.

**Example 1: LSAM.** We consider the case of LSAM, in which only the last layer is perturbed to construct the phantom parameter. Since this is equivalent to considering the representation function $\phi$ to be a constant, we only need to compute the feature gradient norms with respect to the last layer, $w$. We have,

$$\|\nabla_w f(\theta)\| = \|\phi_\theta(x)\|. \tag{25}$$

This implies that the log importance re-weighting factor $\tilde{\lambda}/\lambda$ is proportional to the norm of the representation $\phi_\theta(x)$.

**Example 2: Two-layer linear network.** We consider the case of a two-layer linear network, with a single hidden layer. Let $f(\theta) = v^\top W x$ where $\theta = (v, W)$. For convenience, we will compute the squared norm of the feature gradient,

$$\|\nabla_\theta f(\theta)\|^2 = \|\nabla_v f(\theta)\|^2 + \|\nabla_W f(\theta)\|^2 \tag{26}$$

$$= \|Wx\|^2 + \|v\|^2\|x\|. \tag{27}$$

$$\tag{28}$$

**Example 3: Multi-layer linear network.** We consider the case of a multi-layer linear network. Let $f(\theta) = v^\top W_1 \cdots W_{L-1} x$ where $\theta = (v, W_1, \ldots, W_{L-1})$. For convenience, we will compute the squared norm of the feature gradient,

$$\|\nabla_\theta f(\theta)\|^2 = \|\nabla_v f(\theta)\|^2 + \sum_{i=1}^{L-1} \|\nabla_{W_i} f(\theta)\|^2 \tag{29}$$

$$= \|W_1 \cdots W_{L-1} x\|^2 + \sum_{i=1}^{L-1} \|W_1 \cdots W_{i-1} v\|^2 \|W_{i+1} \cdots W_{L-1} x\|^2 \tag{30}$$

$$= \sum_{j=1}^{L} a_j \|W_j \cdots W_{L-1} x\|^2 \tag{31}$$

where $a_j = \|W_1 \cdots W_{j-1} v\|^2$ is constant with respect to $x$. This means that the squared norm of the feature gradient is proportional to a weighted sum of the squared norms of the representations at each layer.

**Example 4: Multi-layer perceptrons with ReLU.** We can also compute the gradient of non-linear architectures, such as multi-layer perceptrons with ReLU activations. In particular, let $f(\theta) = v^\top \sigma(W_1 \sigma(W_2 \cdots \sigma(W_{L-1} x)))$ be defined by a sequence of layers with the ReLU activation function $\sigma(x) = \max(0, x)$. For a particular input, we can compute the output of a particular layer $i$ as,

$$\sigma_i(x) = \sigma(W_i \sigma(W_{i+1} \cdots \sigma(W_{L-1} x))). \tag{32}$$

observe that for a given input $x$, if we define the matrix $A_i = \mathrm{diag}(\mathbb{I}\{\sigma_i(x) > 0\})$, then observe that the output of the network can be written as,

$$f(\theta) = v^\top A_1 W_1 A_2 W_2 \cdots A_{L-1} W_{L-1} x. \tag{33}$$

For convenience, we will compute the squared norm of the feature gradient,

$$\|\nabla_\theta f(\theta)\|^2 = \|\nabla_v f(\theta)\|^2 + \sum_{i=1}^{L-1} \|\nabla_{W_i} f(\theta)\|^2 \tag{34}$$

$$= \|A_1 W_1 \cdots A_{L-1} W_{L-1}\|^2$$
$$+ \sum_{i=1}^{L-1} \|A_1 W_1 A_2 W_2 \cdots A_{i-1} W_{i-1} v\|^2 \|A_{i+1} W_{i+1} \cdots A_{L-1} W_{L-1} x\|^2 \tag{35}$$

$$= \sum_{j=1}^{L} a_j \|A_j W_j \cdots A_{L-1} W_{L-1} x\|^2 \tag{36}$$

$$= \sum_{j=1}^{L} a_j \|\sigma_j(x)\|^2 \tag{37}$$

$$\tag{38}$$

where $a_j = \|A_1 W_1 \cdots A_{j-1} W_{j-1}\|^2$. Unlike the multi-layer linear case, this constant depends on $x$ through $A_j$. This means that the squared norm of the feature gradient is proportional to a weighted sum of the squared norms of the representations at each layer, but where the weights depend on the input itself. Due to this, a general interpretation of the importance weighting effect is more difficult to obtain.

**General multi-layer neural networks.** In general, mutli-layer neural networks can introduce non-linearities that can make this gradient difficult to analyze. However, we suspect that the computation for mutli-layer linear networks will provide a sensible approximation for the feature gradient norm for multi-layer neural networks.

## E   DATA DISTRIBUTION

In this section, we describe the data distribution for the toy setup.

The data in the toy setup is a concatenation of two features that are independently drawn conditional on a label $y$. Before precisely defining the entire distribution, we will specify the feature distributions individually.

**Easy feature distribution.** We sample the easy feature, conditioned on the label $y \in \{-1, 1\}$, from a latent variable $z$. The latent variable $z$ is sampled from the uniform distribution over the interval $[0, 1]$ if $y = 1$ and from a uniform distribution over the interval $[-1, 0]$ if $y = -1$. The easy feature is then defined as the 2-dimensional vector $[z, z]$, and multiplied by a fixed scale parameter $a_{\text{easy}}$. Together,

$$z_{\text{easy}} \sim \begin{cases} \mathcal{U}(0, 1) & \text{if } y = 1 \\ \mathcal{U}(-1, 0) & \text{if } y = -1 \end{cases} \tag{39}$$

$$x_{\text{easy}} = a_{\text{easy}} \cdot [z_{\text{easy}}, z_{\text{easy}}] \tag{40}$$

**Hard feature distribution.** Similarly, we sample the hard feature, conditioned on the label $y \in \{-1, 1\}$, from a latent variable $z$. The latent variable $z^2$ is sampled from the uniform distribution over the interval $[0, (2\pi\chi/360)^2]$ if $y = 1$ and from a uniform distribution over the interval $[-(2\pi\chi/360)^2, 0]$ if $y = -1$, where $\chi$ is a fixed scalar representing the complexity. The hard feature is then defined as the 2-dimensional vector $[-z \cos z, z \sin z]$, multiplied by a fixed scale parameter $a_{\text{hard}}$, and added to some 2-dimensional uniform noise $\eta \sim \mathcal{U}([0, 0.5] \times [0, 0.5])$. Together,

$$z_{\text{hard}}^2 \sim \begin{cases} \mathcal{U}(0, (2\pi\chi/360)^2) & \text{if } y = 1 \\ \mathcal{U}(-(2\pi\chi/360)^2, 0) & \text{if } y = -1 \end{cases} \tag{41}$$

$$x_{\text{hard}} = a_{\text{hard}} \cdot [-z_{\text{hard}} \cos z_{\text{hard}}, z_{\text{hard}} \sin z_{\text{hard}}] + \eta \tag{42}$$

**Noise.** For the variants that we present in the appendix, we add Gaussian noise $\varepsilon \sim \mathcal{N}(0, \sigma^2)$, feature label noise $u \sim R(p)$, and feature dropout noise $v \sim B(q)$, where $R(p)$ is the Rademacher

distribution with probability $p$ of $-1$ and probability $1-p$ of $1$, and $B(q)$ is the Bernoulli distribution with probability $q$. The noise is added to the easy feature after it is scaled by $a$.

$$x_{\{\text{easy,hard}\}} = \varepsilon + u \cdot v \cdot a_{\{\text{easy,hard}\}} \cdot [z_{\{\text{easy,hard}\}}, z_{\{\text{easy,hard}\}}] \tag{43}$$

In general, we choose $a_{\text{easy}} = 2$ and $a_{\text{hard}} = 0.25$.

## F  DOMAINBED

Table 2: Comparing the domain transfer performance of SAM and SGD on the DomainBed datasets.

| OfficeHome | | | | | PACS | | | | |
|---|---|---|---|---|---|---|---|---|---|
| | Domain | | | | | Domain | | | |
| | A | C | P | R | | A | C | P | S |
| SGD | 0.687 | 0.716 | 0.853 | 0.777 | SGD | 0.910 | 0.897 | 0.955 | 0.893 |
| SAM | **0.709** | **0.745** | **0.858** | **0.784** | SAM | **0.914** | **0.921** | **0.964** | **0.908** |

| VLCS | | | | |
|---|---|---|---|---|
| | Domain | | | |
| | C | L | S | V |
| SGD | 0.993 | 0.736 | 0.748 | 0.812 |
| SAM | **0.996** | **0.748** | **0.776** | **0.830** |

In this section, we present comparison of SAM and SGD on DomainBed (Gulrajani & Lopez-Paz, 2020). DomainBed is a benchmark consisting of datasets with realistic domain shifts. Thus, the results on DomainBed are indicative of SAM's performance under real-world domain shifts in which the features that are useful for in-distribution classification may differ from the features that are useful for out-of-distribution classification. We compare SAM and SGD on three of the DomainBed datasets: OfficeHome, PACS, and VLCS.

### F.1  DATASETS

**OfficeHome** is a dataset consisting of images from four domains: Art (A), Clipart (C), Product (P), and Real-World (R).

**PACS** is a dataset consisting of images from four domains: Art (A), Cartoon (C), Photo (P), and Sketch (S).

**VLCS** is a dataset consisting of images from four domains: Caltech101 (C), LabelMe (L), SUN09 (S), and VOC2007 (V).

We use the standard test/train splits of each of the datasets.

### F.2  EXPERIMENTAL SETUP

To evaluate domain transfer performance for a given dataset and domain, we train a classifier on the training set of all domains except the target domain. Following the setup of the main paper, we extract the quality of the representation by training a linear probe on the training dataset of the target domain.

More precisely, given the representation $\phi$ of the classifier that has been pre-trained on the other domains, we aim to minimize the following loss on the training set of the target domain:

$$\mathcal{L}(u) = \mathbb{E}_{(x,y) \sim D_{\text{train}}} \left[ \ell(u^\top \phi(x), y) \right] \tag{44}$$

where $D_{\text{target}}$ refers to the training distribution of the target domain, $u$ is the linear probe we are optimizing, $\ell$ is cross-entropy loss, and $x$ and $y$ are the input and label of the training example. We evaluate by measuring the accuracy of the linear probe on the test set of the target domain.

We perform a hyperparameter sweep over $\rho \in \{0, 0.03, 0.05, 0.1\}$ and the learning rate $\eta \in \{0.005, 0.01, 0.02\}$. We validate on a small held-out validation set (10% of the training dataset) selected from the training dataset split. We report the results of the best hyperparameter setting on the test set of the target domain.

## F.3 RESULTS

We present the results of SAM and SGD on the three datasets in Table 2. We find that SAM out-performs SGD on all three datasets, confirming that SAM improves the representation quality of the neural network in a variety of domain transfer settings.

## G EXTENDED FIGURES

We provide extended versions of Figures 3 and 5 in Figures 7 and 8, respectively. These figures show the Lorenz curves for the real and phantom importance weights $\lambda_i$ and $\tilde{\lambda}_i$ and the median importance weighting as a function of the contribution of each feature, respectively. We plot each result over multiple checkpoints over the duration of training.

**Results.** We observe that the trends discussed in Section 6.2 are consistent across the duration of training. The phantom importance weights $\tilde{\lambda}_i$ are more uniformly distributed than the real importance weights $\lambda_i$. Further, the median importance weighting is higher for points with a higher contribution of the easy feature than the hard feature.

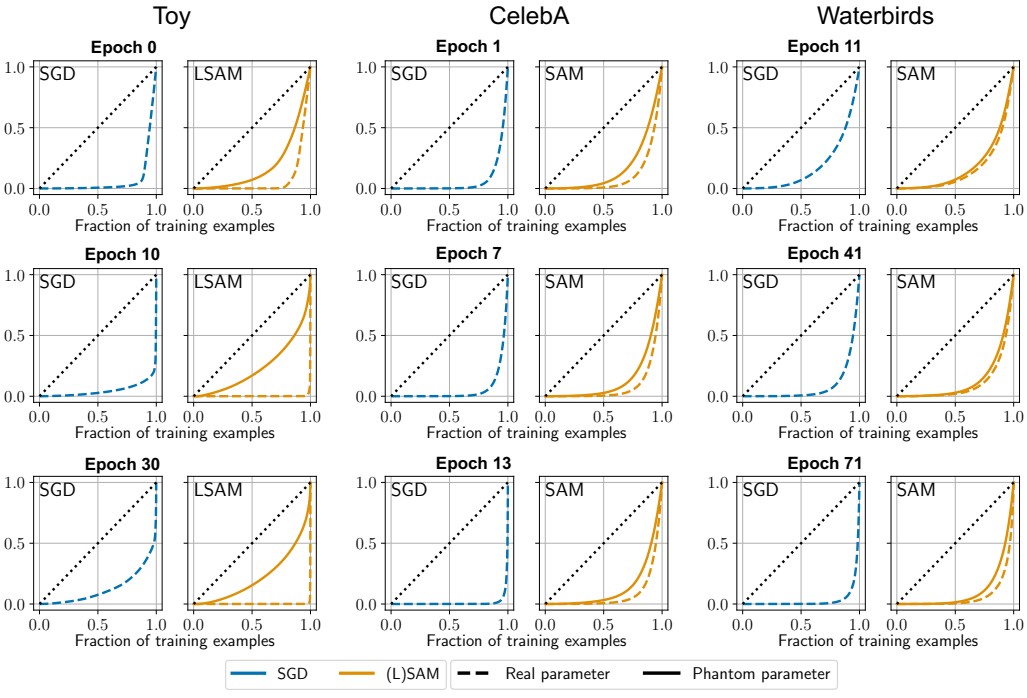

Figure 7: Extended version of Figure 3. Lorenz curves for the real and phantom importance weight $\lambda_i$ and $\tilde{\lambda}_i$. The dotted diagonal line represents the Lorenz curve for a uniform distribution. The closer this curve is to this diagonal, the more equally the importance weights are spread. In blue, we plot the Lorenz curves for each SGD checkpoint. In orange, we plot the Lorenz curves for each LSAM checkpoint. The update step gradient is computed at real parameter for SGD, and the phantom parameter for SAM. We include the curves for the toy (left), CelebA (center), and Waterbirds (right). We observe that within the SAM checkpoints, the weights of points evaluated at the phantom parameter are closer to uniform than when evaluate at the real parameter. Further, in comparison to an SGD checkpoint, the phantom parameter of SAM weights points more uniformly. We plot each result over multiple checkpoints over the duration of training (epoch in bold).

## H VARIANTS OF THE TOY SETUP

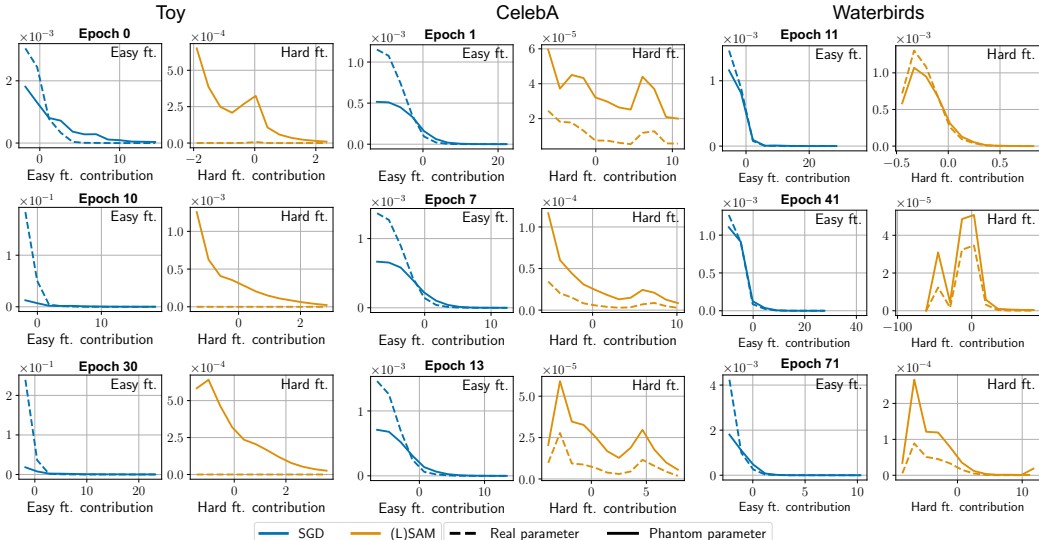

Figure 8: Extended version of Figure 5. Median importance weighting as a function of the contribution of each feature. We partition the data into bins based on the contribution of the easy and hard features ($yv_{\mathsf{easy}}\Phi_{\mathsf{easy}}$ and $yv_{\mathsf{easy}}\Phi_{\mathsf{hard}}$), as defined in Section 6.2. For each of these bins, we plot the median importance weight term $\lambda_i$ for the points in the bin. We include the corresponding plots for the toy (top), CelebA (center), and Waterbirds (bottom). We plot each result over multiple checkpoints over the duration of training (epoch in bold).

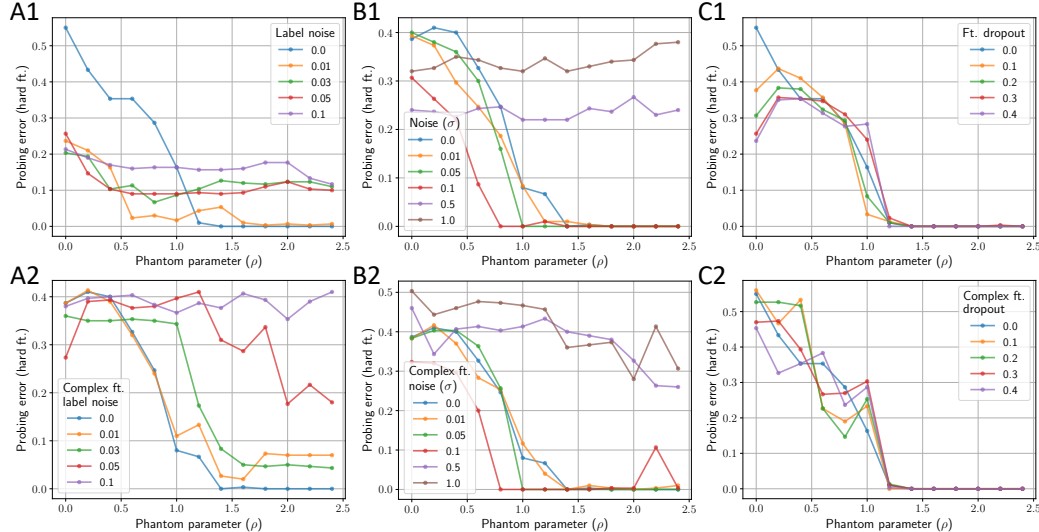

Figure 9: Variants of the toy setup to include noise in the feature distributions (compare with Figure 2). We include three types of noise distributions: (**A**) label noise, (**B**) Gaussian noise, and (**C**) feature dropout. For each type of noise, we apply the noise to: (**1**) both the simple and complex features, and (**2**) only the complex feature. We plot the probing error of the hard feature as a function of the LSAM phantom parameter $\rho$. Note that $\rho = 0$ corresponds to the baseline SGD model.

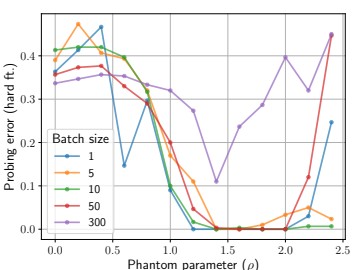

Figure 10: Variant of the toy setup in which we vary the batch size (compare with Figure 2). We plot the probing error of the hard feature as a function of the LSAM phantom parameter $\rho$. Note that $\rho = 0$ corresponds to the baseline SGD model.

In this section, we present variants of the toy setup to include noise in the feature distributions and to vary batch size. In general, our results are consistent with the results from the main paper. We observe that LSAM is robust to a wide range of noise distributions and batch sizes.

## H.1 NOISE DISTRIBUTIONS

In order to explore how LSAM behaves when features include noise, we train classifiers using the toy setup with three types of noise distributions.

**Label noise.** We add label noise by randomly flipping the labels of a fraction of the features when generating the training, as defined precisely in Section E. Note that this is different from how label noise is typically defined: we are not flipping the labels of the training examples, but rather the labels of the features. We vary the fraction of label noise from 0% to 10%.

**Gaussian noise.** We add Gaussian noise to the features, as defined precisely in Section E. We vary the standard deviation of the Gaussian noise from 0 to 1.0.

**Feature dropout.** We add feature dropout to the features, in which, with some probability, we replace features with 0, as defined precisely in Section E. We vary the probability of dropout from 0 to 0.4.

**Hard-feature-only noise.** For each type of noise, we also consider a variant in which the noise is only added to the hard feature.

In general, we find that LSAM improves the probing error of the hard feature in comparison to SGD for all types of noise (Figure 9). This suggests that the mechanisms of LSAM that improve feature representation quality are robust to noise in the feature distributions.

## H.2 BATCH SIZE

We also vary the batch size in the toy setup. We plot the probing error of the hard feature as a function of the LSAM phantom parameter $\rho$ in Figure 10. Note that $\rho = 0$ corresponds to the baseline SGD model. We observe that LSAM is generally robust to a wide range of batch sizes, and that the optimal value of $\rho$ is similar across batch sizes. However, as batch size increases to be very large, the improvement of LSAM degrades, consistent with [CITE]. In addition, we observe that for large $\rho$, the probing error of the hard feature is more sensitive to batch size.

