# OpenReview forum: "Sharpness-Aware Minimization Enhances Feature Quality via Balanced Learning"
_ICLR.cc/2024/Conference — ICLR 2024 poster_

### Official Review · Reviewer_Jhr2 · 2023-10-27

**Soundness:** 3 good
**Presentation:** 3 good
**Contribution:** 3 good
**Rating:** 6
**Confidence:** 3

**Summary:**

This paper shows that SAM can enhance the quality of features in datasets containing redundant or spurious features and this effect can't be explained by in-distribution alone. They show two possible mechanisms for this effect, namely (1) SAM will upweight examples that can't predict well using so-called 'hard features', and (2) the modified gradient in SAM optimizer will attribute more to 'hard features' instead of well-learned 'easy features'. They support the claims with a simple controlled toy experiment and also experiment on real datasets.

**Strengths:**

* **Originality.** To the reviewer's knowledge, this paper is very novel. The phenomenon that SAM may learn more diverse features than SGD even when both reach nearly perfect generalization hasn't been reported before.

* **Quality.** The experiments conducted are complete and thorough.

* **Significance.** The reviewer believes that the finding will deepen the understanding of the mechanism of SAM.

**Weaknesses:**

* **Clarity.** The reviewer feels that clarity requires improvement in the paper. For example, the legend of Figure 5 is hard to follow. Also, the equality (1) seems to contain a typo.

* **Verification of mechanism.** A possible way to verify whether the effect pointed out in the paper really contributes to the phenomenon is to try designing different variants using the discovered mechanism. The paper now only verifies that the importance weighting mechanism exists in real-world experiments but hasn't connected it with the diversity of features.

**Questions:**

* **Q1.** Is it possible to verify the effect of the found mechanisms? (see weaknesses)

* **Q2.** Can the authors try disentangle the two mechanisms and verify which is more important for the phenomenon?

---

> ### Author Response · Authors · 2023-11-17
>
> Thank you for your thoughtful review, we appreciate your feedback!
>
> > The reviewer feels that clarity requires improvement in the paper. For example, the legend of Figure 5 is hard to follow. Also, the equality (1) seems to contain a typo.
>
> Thanks for pointing this out. We will address all issues of clarity in the revised manuscript. For reference here, would like to clarify the meaning of the legend of Figure 5:
>
> As a preliminary, Figure 5 aims to show that points that are well fit by the easy feature are up-weighted, and points that are poorly fit by the complex feature are down-weighted, on average. As a measure of how well each point is fit by the easy feature, we define the “signed easy feature component” as $y v_{\text{easy}} \Phi_{\text{easy}}$. This term represents the magnitude of the output of the easy feature component of the network signed by whether the prediction is correct. When this component is large (well fit by easy feature), we observe that points are generally up-weighted. When the corresponding component for the hard feature is small (poorly fit by hard feature), we observe that points are generally up-weighted.
>
> In the figure caption, we will emphasize that we are plotting, on the x-axis, how well each feature is fit, and on the y-axis, how much the corresponding point is weighted (on average).
>
> > Q1. Is it possible to verify the effect of the found mechanisms?
>
> > Q2. Can the authors try disentangle the two mechanisms and verify which is more important for the phenomenon?
>
> We agree with this concern, and we present this as a limitation in our conclusion section. We want to clarify the challenges associated with verifying and quantifying the effects in practice. In particular, we want to explain the challenges in establishing a causal relationship between each effect and the improvement in feature diversity. Further, we argue that this challenge is precisely why we must rely on a toy model for understanding.
>
> For the toy example, we are able to determine that both effects cause improvement in feature quality through the experiment presented in Figure 6. The exact experiment is explained precisely in Appendix C, but we summarize here. To isolate the learning rate effect, we train with SGD, but where the feature gradients term—associated with the learning rate effect—is computed with artificially balanced features.
>
> Ideally, we would like to perform the same experiment on realistic data. However, in order to artificially balance features, we would have to understand precisely how each feature is represented throughout the neural network, in order to make the appropriate perturbation. This is challenging because the community does not currently understand how features are represented throughout the network.
>
> We can approximate this effect by taking advantage of our observation that a single SAM step will balance features. We can compute the feature gradients term under a SAM perturbation, which approximates the term if features were balanced. However, when training with SAM normally, the benefits of both the importance weighting effect and the feature learning effect are compounded. By approximating only one of these effects at every step means that we cannot take advantage of the cumulative balancing step of both terms, and thus such an experiment will underestimate the effect of each term individually.
>
> Due to the challenge of directly verifying the mechanisms on real data, we need an alternative way to verify our hypothesis. This is precisely the reason that establishing an interpretable toy setup that models the behavior of SAM is an important tool for understanding.

---

> > ### Comment · Reviewer_Jhr2 · 2023-11-23
> >
> > Thank the authors for the reply. I will keep my score.

---

### Official Review · Reviewer_qpzp · 2023-11-01

**Soundness:** 2 fair
**Presentation:** 1 poor
**Contribution:** 2 fair
**Rating:** 6
**Confidence:** 4

**Summary:**

The paper focuses on the effectiveness of Sharpness-Aware Minimization (SAM).
The authors claims that SAM can enhance downstream generalization through feature diversity.
Empirical analysis of SAM trues to demonstrate that SAM offers feature-diversifying benefit.

**Strengths:**

- The paper provides clear motivation.
- The paper provides several experimental results from various perspectives.

**Weaknesses:**

- Although the paper introduces several concepts including feature-probing error or phantom parameter, but it is still unclear why SAM can improve feature diversity.
- What is the meaning of the ratio $v_{\text{hard}}/v_{\text{easy}}$? Why we can say the high ratio means re-balancing effect?
- Why do we need to perturb the last layer only? It would be nice to describe in detail.

**Questions:**

See Weaknesses.

---

> ### Author Response · Authors · 2023-11-17
>
> Thank you for your thoughtful review, we appreciate your feedback!
>
> We appreciate the suggestions to improve clarity and we hope to address them below. In our revised version, we will address each point explicitly in the paper.
>
> > Although the paper introduces several concepts including feature-probing error or phantom parameter, but it is still unclear why SAM can improve feature diversity.
>
> We’d like to ask for further clarification on what you mean that “it is still unclear why SAM can improve feature diversity”.
> If you are concerned with our definition, we define feature diversity to measure how well each algorithm can achieve small linear probing error for all available features. More formally, we define a measure of feature diversity as the worst case (maximal) ProErr (see Equation 1) over the features. The lower the error, the more diverse the features. We will certainly clarify this precisely in the revisions.
> If you are concerned with the methodology of the paper, please let us know if there is any particular issue with our argument: in general, we construct a toy setup where we show that when multiple features are present, SAM can promote the learning of features that SGD performs poorly on. We confirm that SAM improves feature diversity on real-world datasets, and we validate our conclusions from the toy setup.
>
> > What is the meaning of the ratio $v_{\text{hard}} / v_{\text{easy}}$?
>
> The quantities $v_{\text{hard}}$ and $v_{\text{easy}}$ refer to the last-layer parameters of the architecture (specified in Equation 4 and following text). They affect the gradient—and therefore update step—of the model by placing weight on the hard and easy features, respectively, as described in Equation 6. Thus, the ratio $v_{\text{hard}} / v_{\text{easy}}$ is important because it determines the relative weight placed on the hard feature compared to the easy feature during the update step. For SGD, the easy feature tends to dominate the update step, and thus by placing more weight on the hard feature will “balance” the feature contributions. This corresponds with increasing the ratio $v_{\text{hard}} / v_{\text{easy}}$.
>
> > Why do we need to perturb the last layer only?
>
> The toy setup (LSAM) uses a last-layer-only perturbation for simplicity and for interpretability. Despite its simplicity, LSAM exhibits many of the core behaviors of SAM, such as improving feature diversity via the balancing effect. In addition to presenting an understanding of LSAM in the toy setup, we verify that the conclusions we make from the toy setup generalize to SAM---where we perturb every layer---on real data. Please refer to Section 6.3, “Verification on Real-world datasets”.

---

> > ### Comment · Reviewer_qpzp · 2023-11-22
> > **Thanks for the kind reponse.**
> >
> > I have read the author’s response, and authors address my concerns in terms of the methodology. I have raised my score to 6, but I think there is still room for improvement in presentation.

---

### Official Review · Reviewer_KbQq · 2023-11-01

**Soundness:** 3 good
**Presentation:** 3 good
**Contribution:** 3 good
**Rating:** 8
**Confidence:** 3

**Summary:**

Sharpness-aware minimization (SAM) is an alternative method to the traditional stochastic gradient descent (SGD) for training neural networks. The core idea behind SAM is to guide models towards "flatter" minima, believed to enhance generalization. However, recent works revealed that flatness is not the only reason for good generalization. This study proposes an additional advantage of SAM: it improves the quality of dataset features, especially in datasets with redundant or unnecessary features. SAM achieves this by suppressing already well-learned features, allowing lesser-known features to be recognized. This process leads to a more balanced model feature update. Consequently, SAM enhances representation learning by promoting feature diversity. This sheds light on SAM's effectiveness in out-of-distribution performance, offering a new perspective on SAM's benefits.

**Strengths:**

Generally, I think this paper provides an interesting view of SAM.

- This paper provides a new point of view that the SAM algorithm can boost feature diversity. In detail, SGD may be biased towards learning the strongest features from the training data, while SAM promotes the learning of a diverse set of features. Such an explanation is different from the view of the sharpness-based or Hessian-based analysis.

- The new view indicates that the SAM can not only benefit in-distribution tasks but also downstream or out-of-distribution tasks. The empirical study supports their results. The findings may inspire the community to develop new algorithms to improve feature learning.

- This paper is well-organized and presented with comprehensive experiments and interesting illustration figures.

**Weaknesses:**

- Incomplete Understanding of Mechanisms: The discovery that SAM can result in a variety of features is interesting. However, the underlying mechanisms of this finding are not entirely comprehended, particularly in the theoretical aspects.

- Clarity in Data Model Description: The study explores SAM from a feature learning perspective. Nevertheless, the data model described in the paper lacks clarity. It would enhance the paper's comprehensibility if the authors could explicitly present the formula of the toy model (perhaps in the Appendix).

- Elaboration on Feature Measurement: The paper introduces a method to measure strong and weak features. I think it would be an independent significant contribution. An in-depth elaboration on this measure would further strengthen the paper's impact.

**Questions:**

- In experiments such as CIFAR-MINIST, do all the data have a strong feature and a weak feature, or only some of the data have a strong feature while others do not? In equation 6, the authors divide the gradient into the direction of the easy feature and the hard feature, so it seems that each data has both strong and weak features.

- A recent theoretical paper [1] shows that SAM can prevent noise memorization by deactivating the neurons that align with noise and, therefore, can get better in-distribution performance. The authors provide a novel view of the strong-weak features but haven't considered the noise in the model. So, I am wondering how the noises (which are independent of the label ) influence the SAM and SGD's out-of-distribution performance.

-  The batch size $m$ of SAM updates plays an important role [2, 3, 4]. It is observed and proved that m-SAM (SAM with batch size $m$) can have different implicit biases and, therefore, get different performances. It would be better if the author could include the batch size in the methodology and have some discussions.

- I checked the appendix in detail and found that the authors calculate the ratio of importance weighting terms for different neural network architectures. The authors only calculate the gradient for deep linear networks and leave the general networks blank. I agree that the general network's gradient is complex and hard to compute. However, adding some discussion for the two-layer neural networks can help the authors better understand the intuition of this paper,  which is still simple and easy to calculate [1, 4].

[1] Chen et al. "Why Does Sharpness-Aware Minimization Generalize Better Than SGD?" arXiv preprint arXiv:2310.07269, 2023.

[2] Foret et al. "Sharpness-aware minimization for efficiently improving generalization." In ICLR, 2021.

[3]  Andriushchenko et al. "Towards understanding sharpness-aware minimization." In International Conference on Machine Learning, pp. 639–668. PMLR, 2022.

[4] Wen et al. "Sharpness minimization algorithms do not only minimize sharpness to achieve better generalization." arXiv preprint arXiv:2307.11007, 2023.

---

> ### Author Response · Authors · 2023-11-17
>
> Thank you for your thoughtful review, we appreciate your feedback!
>
> > It would enhance the paper's comprehensibility if the authors could explicitly present the formula of the toy model.
>
> Of course! We have added this to the revised paper. Please see Appendix E in the revised copy.
>
> > An in-depth elaboration on [the linear probing error] measure would further strengthen the paper's impact.
>
> We will add discussion of this measure in detail in the appendix in the form of extended related work, in which we will discuss the following. The linear probing measure—in which we measure feature quality by measuring the error of the linear classifier on top of the representation layer that minimizes error of the desired feature—has been applied previously.  Previous work has shown that the last layer representations often encode features that are useful for out-of-distribution generalization even when the neural network performs poorly OOD (without linear probing) [1]. Surprisingly, linear probing error is often close to the error as if the neural network was finetuned (with all parameters) on the downstream data [2], which makes it a strong measure of feature quality. In addition, linear probing is often itself used for downstream generalization when there is only limited downstream data available, or in compute constrained environments, notably for [3].
>
> > In experiments such as CIFAR-MINIST, do all the data have a strong feature and a weak feature, or only some of the data have a strong feature while others do not?
>
> In all cases, every data point has both features: for example, in CIFAR-MNIST, the “hard” CIFAR feature was {truck, airplane}, and an image of a truck or an image of an airplane would be present, and similarly for MNIST, {0, 1}. This is also true of every other dataset that we consider.
>
> We have added experiments to the toy setting but where each feature is not necessarily present in every training example. Consistent with our results, we find that in this setting, SAM improves performance. Please see our “General Response” for specific details.
>
> >  So, I am wondering how the noises (which are independent of the label ) influence the SAM and SGD's out-of-distribution performance.
>
> > It would be better if the author could include the batch size in the methodology and have some discussions.
>
> We have added experiments to test various types of noise and varied batch size in our toy setting. Consistent with our results, we find that in both of these settings, SAM improves performance. When we vary batch size, consistent with prior work, small batch size tends to improve performance with SAM. Please see our “General Response” for specific details.
>
> > Adding some discussion for the two-layer neural networks can help the authors better understand the intuition of this paper.
>
> We have added the analysis of the importance weighting ratio for a multi-layer ReLU network (MLP) to the revised paper (Appendix D. Example 4). Similar to the multi-layer linear case, the importance weighting ratio is a weighted sum of the intermediate layer outputs. However, unlike the multi-layer linear case, these weights can depend on the inputs.
>
> Please let us know if you have any additional questions or want further clarification!
>
> [1] https://openreview.net/forum?id=Zb6c8A-Fghk#
>
> [2] https://arxiv.org/abs/2202.06856
>
> [3] https://arxiv.org/abs/2212.07143

---

> ### Comment · Reviewer_KbQq · 2023-12-05
>
> I have read the author's rebuttal and appreciate their efforts in addressing most of my concerns about the presentation. In light of their responses, I am inclined to revise my evaluation, raising the score to an 8. However, my revised score is a conservative estimate, oscillating between 6 and 8 due to the lack of theoretical understanding. Additionally, for greater transparency and clarity, I strongly encourage the authors to include their code's release in the paper's final version and polish their appendix.

---

### Official Review · Reviewer_BgWc · 2023-11-01

**Soundness:** 2 fair
**Presentation:** 2 fair
**Contribution:** 3 good
**Rating:** 6
**Confidence:** 3

**Summary:**

This paper empirically studies the mechanism of Sharpness-Aware Minimization (SAM), which improves generalization beyond in-distribution data (e.g., on downstream or out-of-distribution tasks). The authors demonstrate that SAM has a significant impact on improving representation quality through feature diversity, i.e., promoting a neural network to learn both easy and hard features. Through toy experiments, the authors isolate two core mechanisms within SAM that lead to better feature quality: 1) importance sampling, and 2) learning rate scaling. The former one is further verified on real-world datasets.

**Strengths:**

- Explaining the effectiveness of SAM from the perspective of feature quality is novel.
- The authors identify that SAM’s feature improvement cannot be solely explained by in-distribution improvements.
- The authors carefully design a toy experiment, based on which two reasonable mechanisms are isolated to explain why SAM can learn the hard-to-learn features.

**Weaknesses:**

- The authors argue that due to the ability to learn diverse features (both easy and hard), SAM can improve the generalizability beyond in-distribution. However, this raises a question: are all these features beneficial to the out-of-distribution generalization or downstream tasks? For example, in Table 1, the hard-to-learn feature in CelebA is the "gender feature" while the task is to predict the "hair color". In this case, the easy features are correctly related to the labels, and there exists spurious-correlation between the hard "gender feature" and the labels. The SAM indeed decreases both the probe errors for easy and hard features, but the latter is harmful for ood generalization. Thus, it seems contradictive between "SAM learns diversity features" and "SAM improved generalizability beyond in-distribution". The authors should provide more explanations.
- While the authors identify two underlying effects to explain why SAM learns diversity features with better quality, the learning rate effect is hard to verify in real-world datasets.
- Minor: In Sec3.1, Eq 1, replace the $\phi_\theta$ to $\Phi_\theta$.

**Questions:**

Please see weaknesses.

---

> ### Author Response · Authors · 2023-11-17
>
> Thank you for your thoughtful review, we appreciate your feedback!
>
> > Are all these features beneficial to the out-of-distribution generalization or downstream tasks?
>
> > [...] It seems contradictive between "SAM learns diversity features" and "SAM improved generalizability beyond in-distribution".
>
> We apologize for any confusion and clarify that indeed feature diversity is beneficial to OOD/downstream tasks. There is no contradiction here. In fact, feature diversity is key for better OOD/downstream performance. We explain why below.
>
> Prior works that study Waterbirds and CelebA typically treat these settings as having a  “core/good” feature (e.g., hair) and a “spurious/bad” feature (e.g., gender) which is merely correlated with the label. These works only care about distribution shifts where the spurious feature in particular is no longer predictive of the label. However, our work takes a more general view where we want to tackle distributions where either one of the features (hair or gender) could possibly become unpredictive of the label.  Thus, we argue that ideally we want our learned representation to have both the “hair” and “gender” feature: this is because a downstream task may require using any one of these features (either hair or gender). If both features are learned, then a linear probe can successfully pick up whichever feature is relevant downstream. We clarify that this does not contradict the “spurious correlation” literature, where the nomenclature can be somewhat misleading: there, the practice is to call the easier feature the “spurious feature” since their interest is specifically in learning the feature that SGD cannot learn (the non-spurious one). However, in the real-world, where any feature (either the easy or the hard one) can be “spurious” in that its correlation with the label may disappear in the downstream task.
>
> To confirm that the features learned are useful for OOD/downstream tasks, we have added additional experiments. We show that SAM improves domain-transfer performance on DomainBed, a realistic dataset of domain shifts. Please see our “General Response” for more specific details about both additions.
>
>
> > While the authors identify two underlying effects to explain why SAM learns diversity features with better quality, the learning rate effect is hard to verify in real-world datasets
>
> We agree with the concern that it is still unclear how the learning rate effect generalizes to a realistic setup. We present this as a limitation in our conclusion section. The key challenge in achieving a similar observation of the learning-rate effect on real data is understanding precisely how features are represented. Since SAM perturbs the weights at every layer, we would need to understand how each feature is represented at every layer of the neural network. Based on the general principle that SAM perturbs weights as a function of how much they affect the output, we suspect that well-learned features will be inhibited by SAM at every layer.
>
> Nonetheless, due to the challenge of directly verifying the mechanisms on real data, we need an alternative way to verify our hypothesis. This is precisely the reason that establishing an interpretable toy setup where the latent features are explicitly disentangled—here we can understand the behavior of SAM.
>
> Please let us know if you have any additional questions or want further clarification!

---

> > ### Comment · Reviewer_BgWc · 2023-11-22
> >
> > Thanks for the clarification. I keep my score after reading the authors’ response and other reviewers' comments.

---

### Official Review · Reviewer_NKN8 · 2023-11-03

**Soundness:** 3 good
**Presentation:** 3 good
**Contribution:** 2 fair
**Rating:** 6
**Confidence:** 4

**Summary:**

This paper studies why Sharpness-Aware Minimization (SAM) can generalize well. They provide an explanation based on the weight that SAM/SGD put on different features, explaining SAM can up-weight features that are harder to learn.

**Strengths:**

I think the idea of balancing features is interesting, and it makes sense to study SAM from this perspective. I agree that sharpness of the loss cannot alone explain the differences of SAM/SGD, and I appreciate this work's attempt to go beyond.

**Weaknesses:**

The paper is inherently more on the empirical side, which I think is fine. However, I think some simplified theory could increase the contribution of the work. Based on this, I'd rate the paper as borderline accept.

**Questions:**

-

---

> ### Author Response · Authors · 2023-11-17
>
> Thank you for your support of our contributions. We are excited to present work that moves beyond the notion that sharpness can explain generalization. We hope that our empirical but principled insights can serve as useful guidance for future theoretical characterizations of SAM, and can motivate the development of principled algorithms.

---

### Official Review · Reviewer_LFgC · 2023-11-06

**Soundness:** 2 fair
**Presentation:** 3 good
**Contribution:** 3 good
**Rating:** 6
**Confidence:** 4

**Summary:**

Several learning theories, e.g., PAC Bayes, explain that flatter minima generalize better.
Thus, SAM is formulted to find flatter minima, and is known to improve in-distribution
generalization.
However, the beyond in-distribution performance is not well understood.
This paper focuses on  a feature-diversifying effect of SAM that is relevant for downstream generalization.
As seen in simplicity bias or gradient starvation,SGD exhibits a bias toward learning simple/easy-to-fit feature, even if the training data has multiple latent features that are all useful for prediction.
In contrust, this paper explans that SAM also learn the hard-to-fit features.
The authors shows this property of SAM by constructing a toy setting consisting of an easy-to-fit and a hard-to-fit features.

**Strengths:**

This paper seems to be the first to show the relationship between easy-to-fit and hard-to-fit features and learning by SAM.
This perspective is useful in that it opens up the possibility that SAMs can solve problems that SGD has, such as gradient starvation.

**Weaknesses:**

While the analysis of the toy setting is interesting, it is unclear to what extent the analysis is applicable to learning with real images such as the winter bird and CelebA datasets.

**Questions:**

Q1.What does it take in the future to extend the theory from toy setting to reality image identification? Please explain how difficult it would be.
Q2: As a question related to Q1, in the toy setting, easy-to-fit and hard-to-fit are separated as latent features, but this may not necessarily be the case in real-world data. For example, is it possible that at the time of the feature extractor, the SGD is trained in such a way that the signals of hard-to-fit features do not reach the output layer in the first place? Is there any way to know that easy-to-fit and hard-to-fit features are disentangled in latent features in a real-world problem?
Q3: How much is the influence of the terms after the second order in Eq.(20), which can be ignored since the hessian trace is small for the purpose of finding the fat minima? In this case, what is the effect during the learning process?
Q4. The word "diversity" is used in this paper; however, it actually refers only to two characteristics, easy-to-fit and hard-to-fit.
In terms of beyond-in-distribution setting,  transferabiliity is also important. For example, robust/non-robust features in the analysis of adversarial examples, etc. could be considered. Please explain the relationship with other features.
Q5. There are oftern noisy features as well as easy-to-fit and hard-to-fit features in real images. Is it possible to analyze the effect of SAM on noisy features that are typically hard-to-fit.

---

> ### Author Response · Authors · 2023-11-17
>
> We agree that there is currently a gap between the assumptions of the toy setting and a more realistic setting. We hope to clarify the specific challenges that we face in generalizing our results to more realistic settings. However, we want to emphasize that we do not require all of these assumptions to be held strictly, and we observe improvements in SAM even when these assumptions are broken. All discussed below:
>
> > It is unclear to what extent the analysis is applicable to learning with real images
> > What does it take in the future to extend the theory from toy setting to reality image identification?
> > Is it possible that the signals do not reach the output layer in the first place.
>
> We present analysis to verify the learning rate effect in real data in Section 6.3. To summarize, we verify the two main predictions from the toy: (1) the distribution over importance weights becomes more uniform under SAM (Figure 4), and (2) SAM up-weights points that are well-fit by the easy feature and poorly fit by the complex feature (Figure 5).
>
> We agree with the concern that it is still unclear how the learning rate effect—in which we argue that SAM balances the step size in the direction of each feature—generalizes to a realistic setup. The key challenge in achieving a similar observation of the learning-rate effect on real data is understanding precisely how features are represented. Since SAM perturbs the weights at every layer, we would need to understand how each feature is represented at every layer of the neural network. Based on the general principle that SAM perturbs weights as a function of how much they affect the output, we suspect that well-learned features will be inhibited by SAM at every layer. We expect that if the signal of a particular feature does not reach the output layer, SAM will promote this feature.
>
> This challenge is precisely the reason that establishing an interpretable toy setup is an important tool for understanding.
>
> > In the toy setting, easy-to-fit and hard-to-fit are separated as latent features, but this may not necessarily be the case in real-world data
>
> We agree, as you mention, that features may not be entirely disentangled in the latent space. Nonetheless, we observe the benefits of feature learning with SAM and can identify the importance weighting effect with realistic data. This suggests that our analysis holds even when the features are not entirely separated as latent features.
>
> > Is there any way to know that easy-to-fit and hard-to-fit features are disentangled in latent features in a real-world problem.
>
> As mentioned above, the community does not have a solid understanding of how features are represented by neural networks. Thus it may be difficult, in general, to determine if features are disentangled. Recent work suggests that features are often at least partially disentangled (see Appendix B.4 of [1]).
>
> However, We suspect, for the reasons enumerated above, that SAM should promote harder features even when they are not entirely disentangled.
>
> > How much is the influence of the terms after the second order in Eq.(20), which can be ignored since the hessian trace is small for the purpose of finding the fat minima?
>
> Exactly as you mention, since SAM is optimizing for minima with a small spectral norm and since $\rho$ is relatively small (in practice, we find that performance is typically maximized for $\rho$ around 0.05 or so), we suspect that the second order term is unlikely to contribute meaningfully to the dynamics.
>
>  > The word "diversity" is used in this paper; however, it actually refers only to two characteristics, easy-to-fit and hard-to-fit. [...] For example, robust/non-robust features.
>
> We call a feature “easy-to-fit” or “hard-to-fit” relative to another depending on which feature SGD learns better. In any real-world scenario, such preferential learning is bound to happen in the presence of multiple features. In your robustness example, non-robust features are known to be preferentially learned by SGD over robust features [2], thus our insights would apply there too. Consistent with this, SAM has been shown to improve adversarial robustness [3].  Besides, our results on DomainBed show that SAM indeed helps learn more transferable features in a realistic setting.
>
> > Is it possible to analyze the effect of SAM on noisy features that are typically hard-to-fit.
>
> Great point! We have added new experiments based on this suggestion. We find that SAM improves domain-transfer performance on DomainBed, a realistic dataset of domain shifts. In addition, we have added experiments that explore variants of the toy in which we add various types of noise to the features. Consistent with our results, we find that LSAM improves performance in this setting. Please see our “General Response” for more specific details about both additions.
>
>
> [1] https://openreview.net/pdf?id=Zb6c8A-Fghk
>
> [2] https://arxiv.org/abs/2006.07710
>
> [3] https://arxiv.org/abs/2305.05392

---

> > ### Comment · Reviewer_LFgC · 2023-11-22
> >
> > Thanks for answering many questions.
> > I wanted to raise the score, however this time I found that the score 6 is next to 8 in the system.
> > It still seems to me that there is a gap between the theory and the real data setting if I were to raise the score to 8.
> > Therefore, I would like to keep the score at 6 but note that it is more positive than 6.

---

### Author Response · Authors · 2023-11-17
**General Response**

We thank the reviewers for their feedback and for their encouragement of our feature diversity perspective on SAM. We are indeed excited about the departure from flatness in explaining SAM’s benefits, and also the potential of our findings to inspire better algorithms for feature learning.

We want to address two concerns/suggestions that were brought up by the reviewers.

We thank the reviewers for their suggestion on studying SAM in the presence of noise, varying batch size and considering the case where not all data points have all the predictive features. We have added all of this in the paper, and summarize the main findings below.

**Toy simulations under different assumptions**

*Adding noise.* We have added variants to Appendix G where we explore the role of label noise and gaussian noise in both features, and in only the harder feature. Consistent with our previous findings, we find that LSAM still improves the quality of the hard feature in the representation (measured by a linear probe) even if it is moderately noisy. Please refer to Figure 10.A and 10.B.

*Exploring when features are not always present.* We have added a variant in which, with some probability for each data point, we remove one of the features (setting it to 0). We find that LSAM still improves the hard feature quality in the representation. We include this with the other additional experiments that inject noise into the features of the toy setup in Appendix G, Figure 10.C.

*Changing the batch size.* We have added a variant in which we vary the batch size in Appendix G. We observe, consistent with the prior work mentioned by Reviewer KbQq, that small batch size improves LSAM. Nonetheless, LSAM is relatively robust to different batch sizes. Please refer to Figure 11.

**Additional verification on real-world data**

We agree with the reviewers that our toy setting makes simplifying assumptions that do not hold in practice. Our paper makes different contributions/claims, and we elucidate below what assumptions are required for each claim. We also add another experiment in a standard real-world out-of-distribution setting of DomainBed.

1. *SAM improves feature diversity:* Our high level observation that SAM improves the representation quality holds in general—it does NOT require explicitly disentangled features or fully predictive features. We test this in our newly added experiment on DomainBed. We train on a set of training domains, and test the resultant feature quality by training a linear probe on a different held-out domain. We find that SAM representations perform better, suggesting that the representation captures good quality representations of more predictive features which may be redundant in-distribution, but are useful OOD. We hope our new experiment addresses the reviewers’ concerns on general applicability of our findings.
 We present a detailed description of the experiment in the updated Appendix F. As can be seen in Table 2, SAM improves transfer performance to a held-out domain for all tested DomainBed datasets.
2. *The mechanism for SAM improving feature diversity:* Our next contribution is a more mechanistic and interpretable understanding of how SAM features. In our toy setting, we find two interpretable effects: the importance weighting effect and the learning rate effect. We believe this provides intuition for why SAM might improve feature diversity in general (observation (1)). However, measuring the importance weighting effect (as we interpret) requires explicitly annotated features in the input, but does not require them to be disentangled in the final layer. We show that the importance weighting effect appears in Waterbirds and CelebA. Measuring the learning rate effect (as we interpret) requires explicit disentanglement of features in the layers perturbed by SAM. We design our toy setting such that this is the case, but we agree that this is not true in general, and that is why we are unable to measure learning rate effect on realistic datasets.


Finally, we have addressed the individual questions and concerns of each reviewer in our individual responses below.

Thank you again for your time in reviewing and reading our responses,

Authors

---

### Meta-Review · Area_Chair_NKLY · 2023-12-07

**Metareview:**

This paper studies why Sharpness-Aware Minimization (SAM) can generalize well and can converge towards flatter minima. The authors provide a new perspective to interpret SAM optimizer by arguing that SAM can enhance the quality of features in datasets containing redundant or spurious features. Extensive experiments support their claims.

Strengths:

(1)   SAM is an effective optimizer to seek flat minima. However, its understanding is still open. This work provides an understanding of SAM optimization. Explaining the effectiveness of SAM from the perspective of feature quality is novel and interesting.

(2)   The paper is well-written, and the idea is easy to follow. The experiments are extensive to support their findings.

Weaknesses:

(1)   The assumption in this work is slightly strong. The main analysis is conducted on the carefully designed toy dataset. More experiments on real-world datasets will largely strengthen this work.

After the authors' response and discussion with reviewers, all the concerns are well-solved. All the reviewers agree to accept this work. Thus, I recommend acceptance.

**Justification For Why Not Higher Score:**

N/A

**Justification For Why Not Lower Score:**

N/A

---

### Decision · Program_Chairs · 2024-01-16

Accept (poster)